# Transcutaneous spinal cord stimulation and motor responses in individuals with spinal cord injury: A methodological review

**Clare Taylor**[1]*, **Conor McHugh**[1], **David Mockler**[2], **Conor Minogue**[1], **Richard B. Reilly**[3,4,5], **Neil Fleming**[1]

**1** Department of Anatomy, School of Medicine, Trinity College, The University of Dublin, Dublin, Ireland, **2** John Stearne Medical Library, Trinity Centre for Health Sciences, School of Medicine, St. James's Hospital, Dublin, Ireland, **3** Trinity Centre for Biomedical Engineering, Trinity College, The University of Dublin, Dublin, Ireland, **4** School of Engineering, Trinity College, The University of Dublin, Dublin, Ireland, **5** School of Medicine, Trinity College, The University of Dublin, Dublin, Ireland

* taylorc1@tcd.ie

## Abstract

### Background

Transcutaneous spinal cord stimulation (tSCS) is a non-invasive modality in which electrodes can stimulate spinal circuitries and facilitate a motor response. This review aimed to evaluate the methodology of studies using tSCS to generate motor activity in persons with spinal cord injury (SCI) and to appraise the quality of included trials.

### Methods

A systematic search for studies published until May 2021 was made of the following databases: EMBASE, Medline (Ovid) and Web of Science. Two reviewers independently screened the studies, extracted the data, and evaluated the quality of included trials. The electrical characteristics of stimulation were summarised to allow for comparison across studies. In addition, the surface electromyography (EMG) recording methods were evaluated.

### Results

A total of 3753 articles were initially screened, of which 25 met the criteria for inclusion. Studies were divided into those using tSCS for neurophysiological investigations of reflex responses (n = 9) and therapeutic investigations of motor recovery (n = 16). The overall quality of evidence was deemed to be poor-to-fair (10.5 ± 4.9) based on the Downs and Black Quality Checklist criteria. The electrical characteristics were collated to establish the dosage range across stimulation trials. The methods employed by included studies relating to stimulation parameters and outcome measurement varied extensively, although some trends are beginning to appear in relation to electrode configuration and EMG outcomes.

**Data Availability Statement:** All relevant data are within the manuscript and its Supporting Information files.

**Funding:** This study was supported by the Disruptive Technology Innovation Fund, grant number DT -2018-0128 (RR, NF, CM) The funders had no role in study design, data collection and analysis, decision to publish, or preparation of the manuscript.

**Competing interests:** The authors have declared that no competing interests exist.

## Conclusion

This review outlines the parameters currently employed for tSCS of the cervicothoracic and thoracolumbar regions to produce motor responses. However, to establish standardised procedures for neurophysiological assessments and therapeutic investigations of tSCS, further high-quality investigations are required, ideally utilizing consistent electrophysiological recording methods, and reporting common characteristics of the electrical stimulation administered.

## Introduction

Transcutaneous spinal cord stimulation (tSCS) is a non-invasive form of neuromodulation in which electrodes are placed on the skin and used to stimulate the spinal circuitries via an electrical current [1–3]. It has been proposed that this tool could not only provide us with greater understanding of spinal inter-neuronal functioning but also enhance the rehabilitation potential for people with neurological disorders, such as spinal cord injury (SCI) [2, 4–6]. As this modality is under the relatively early stages of investigation with injured individuals, there is still much to learn about its implementation and clinical potential.

Modelling studies have demonstrated that electrical pulses delivered from spinal cord stimulation (SCS) preferentially depolarize sensory afferents in the posterior roots, which can elicit a motor reflex response [7, 8]. This response has been termed a posterior root-muscle (PRM) reflex [9], multisegmental monosynaptic response (MMR, [5]), or transpinal evoked potential (TEP, [10, 11]), amongst other nomenclature. As an alternative to the H-Reflex, the study of the PRM reflex allows us to expand the neurophysiological assessment of sensory-motor transmission of stimuli to more muscle groups and provide greater insights into the functioning of spinal circuitries [4, 12].

Spinal stimulation via transcutaneous input is believed to be distinguished from direct stimulation of motor efferents, such as in traditional nerve or muscle stimulation techniques, due to the transsynaptic transmission of motor responses via monosynaptic or oligosynaptic pathways [13]. Several studies have investigated the reflex nature of responses, using paired pulses to demonstrate post-activation depression (PAD), in which the amplitude of the second pulse of a pair is attenuated with respect to the first [14–22]. Additionally, the inhibition of tSCS evoked responses via tendon vibration is consistent with the stimulation of reflex responses from Ia afferents [5, 22]. Other studies have focused on alternative methods to demonstrate spinal neuromodulation of motor responses through outcomes such as increased response latencies [2, 23], differential muscle activation patterns [24], phase-dependent modulation of reflex responses [5, 14] and the alteration of amplitudes subsequent to afferent input [25] or interlimb conditioning [26].

It is also theorized that SCS can modulate inter-neuronal spinal excitability, and that this may account for the observed motor recovery when used in individuals with SCI [19, 27–29]. By activating networks such as central pattern generators (CPGs) and the propriospinal system (PSS), spinal excitability may be augmented and the threshold for motor impulse propagation lowered [30, 31]. A CPG is a spinal network of neurons believed to be capable of generating a co-ordinated rhythmic motor output such as locomotion in the absence of input from supraspinal centres and/or afferent feedback [32]. The PSS has been described as an interface between spinal segments that contributes to movement and rhythmic coordination [33, 34], as

well as providing a background of subthreshold excitation [30, 35]. The modulation of spinal networks and altered threshold for impulse propagation may explain the results of several studies using tonic spinal stimulation that have reported improved motor outcomes in chronically paralysed individuals [36–38].

Indeed, in the case of SCI, spinal neuromodulation may provide greater functional recovery beyond the capacity of currently available therapies, particularly after more severe or chronic injury [30, 39]. Thus far, a selection of studies investigating the effects of tSCS on motor rehabilitation in chronic SCI have published cases of improved lower limb [19, 40–42], trunk [37] and upper limb functioning [36, 43–45]. Despite these promising initial results, a recent review evaluating the therapeutic effects of tSCS on motor recovery in individuals with SCI reported that due to the small heterogenous sample sizes, diverse range of outcome measures and low methodological quality of reviewed studies, no conclusions can be drawn on its effectiveness [46].

The most consistently employed measure across studies directly evaluating motor responses from spinal stimulation appears to be surface electromyography (EMG) [46, 47]. EMG can be used either to quantify evoked motor potentials and/or the degree of voluntary muscle activation facilitated by tSCS. In evaluating any clinical therapy, the effects may not be initially present at a functional level but could be present at the neuromuscular level, and these effects may be quantified by EMG [48, 49]. EMG can also estimate the real-time effects of tSCS parameters and application on spinal excitability [17, 22, 50, 51], potentially even providing physiological input for closed-loop control paradigms [52, 53]. While the use of EMG has many advantages, it is subject to limitations that must be carefully considered when recording, processing, and interpreting the data [54]. The use of EMG with neurologically impaired individuals presents many additional challenges [48], such as sub-optimal normalisation procedures [55] and signal contamination by involuntary tonic activity [56]. Furthermore, stimulation artifact can contaminate the EMG motor signals produced by tSCS [49].

Presently, there is a lack of consensus surrounding the standardised use of tSCS to facilitate motor responses in individuals with SCI [46]. Optimal stimulation parameters and experimental protocols remain unclear and there is much variability seen in outcome measurement. The extent of this methodological variability would benefit from a systematic evaluation in order to synthesize the information on currently employed parameters and provide recommendations to enhance the development of future studies investigating the properties and efficacy of tSCS. As such, the objective of this systematic review was to methodologically appraise studies which used tSCS to generate motor activity in persons with SCI. In doing so, this review sought to critique the quality of included trials, review intervention parameters employed and compare the methods of evaluating motor responses with surface EMG.

## Methods

A systematic review of the literature was undertaken using the methodology described by the Preferred Reporting Items for Systematic Review and Meta-Analysis (PRISMA) Protocols Statement [57].

### Search strategy

An extensive literature search was carried out using the following electronic databases: EMBASE, Medline (Ovid) and Web of Science. It included studies published until 31$^{st}$ May 2021. The initial search was kept broad to in an attempt to capture all possible spinal stimulation studies using varying nomenclature. The search was built with the help of a research librarian (DM) based on anchoring terms from the following categories: ***spinal cord stimulation, spinal cord injury and motor responses***. Search terms were expanded using a vast list of

alternative terminologies, truncations, and abbreviations. The exact search algorithm and medical subject heading (MeSH) terms used with each engine are presented in S1 Appendix. Additional relevant publications were also sought out by retrospectively completing a manual search of the bibliographies of all included studies and by manually searching for other publications from authors of tSCS studies that were identified in the search.

### Study selection procedure

Two independent reviewers (CT, CMcH) completed an initial title screen to remove any highly irrelevant papers. The eligibility criteria (Table 1) were designed based on the PICO model (Population, Intervention, Comparison, Outcome). Pilot testing of the exclusion criteria was conducted using a subset of 150 abstracts screened by both reviewers and the reasons for exclusion were documented. The reviewers then completed the abstract screening and a Cohen's Kappa of 0.88 was reached. This correlation was deemed sufficient. Finally, the full texts were reviewed for inclusion and all reasons for exclusion were recorded. If there was any uncertainty about inclusion, a third reviewer (NF) was consulted until a consensus was reached. The independent reviewers were not blinded to the study authors, institutes, or journal titles. As there were a small number of publications meeting the inclusion criteria, we did not require a minimum sample size. The literature search was last performed on the 31st May 2021.

### Quality appraisal

In order to appraise the quality of the included full texts, the Downs and Black (D&B) Checklist was employed [58]. This tool has been used to evaluate non-randomised controlled trials (RCTs) in other systematic reviews pertaining to populations with SCI [59–61] and its use is recommended by the SCIRE (Spinal Cord Injury Research Evidence) Research Team [62]. The D&B Checklist has also been recommended for use in assessing non-RCTs due to its psychometric properties [63, 64].

Two independent reviewers (CT, CMcH) conducted the quality appraisal, and any disagreement was discussed with a third reviewer (NF) until consensus was reached. The D&B

**Table 1. The eligibility criteria to determine suitable studies for inclusion in the full-text systematic review.**

|  | Inclusion | Exclusion |
|---|---|---|
| **Participants** | • Aged > 18 years | • Animal studies |
|  | • A primary diagnosis of spinal cord injury (any level, complete or incomplete). | • Aged < 18 years |
| **Intervention** | • Transcutaneous spinal cord stimulation aimed at producing a motor response. | • Magnetic stimulation or direct current stimulation |
|  | • Pulsed and continuous electrical spinal stimulation protocols. | • Peripheral stimulation such as Functional Electrical Stimulation (FES) or Neuromuscular Electrical Stimulation (NMES) |
|  |  | • Paired Associative Stimulation (PAS) |
|  |  | • Epidural spinal cord stimulation (eSCS). |
| **Comparison** | • No intervention, sham intervention, or pre-post analysis |  |
| **Outcomes** | • A measure of motor activity in a targeted muscle/muscle groups by EMG recordings | • The primary outcome selected and reported on measures pain, autonomic function, or spasticity |
| **Data analysis** | • Study must report details pertaining to the transcutaneous spinal cord stimulation parameters utilised | • Studies that fail to specify any stimulation parameters |
| **Publication type** | • Original primary data from a prospective interventional, quasi-experimental, or observational study | • Review articles, conference proceedings, expert opinions, or any other secondary publication |
|  | • Published in peer reviewed journal until 31st May 2021 |  |
|  | • Published in English | • Abstract or full text not available in English |

Checklist is a 27-item list that evaluates methodological strengths and weaknesses of articles based on the categories of *(1) Reporting, (2) Internal Validity (Bias), (3) Internal Validity (Confounding), (4) External Validity* and *(5) Power* [58]. Power level calculations (1-β error probability) for the checklist were made using the G*Power Application [65] and analysis was derived from the statistical tests applied to the main study findings. The following marks were awarded: 1 point for a power level of 70%, 2 points for power level of 80%, 3 points for power level of 85%, 4 points for power level of 90%, 5 points for power level of 95%. The modified version of the D&B Checklist was not used, as the authors felt it important to adequately represent the sufficient powering of studies as per the original checklist. The following rounded cutoff points were used to categorize studies by quality [66]: excellent (91%–100%), good (71%–90%), fair (51%–70%), and poor (0%–50%).

## Data extraction

Results were generated from data extracted to standardised spreadsheets which included (i) study type, (ii) sample characteristics and clinical variables, (iii) intervention parameters, (iv) outcome measurements (v) EMG data collection and signal processing (vi) and safety/adverse events. Table results were pooled by two study members until consensus was reached, and disagreements were discussed with the third reviewer. Studies investigating similar objectives were grouped together for comparison, in particular, a distinction was made between neurophysiological experiments and therapeutic investigations seeking motor rehabilitation.

The electrical and timing characteristics of the stimulation signals used in tSCS vary widely, making comparisons between studies difficult. Moreover, there is a lack of consistency in the definition of these parameters. This study sought to clearly define key stimulation parameters and descriptors and, where possible, extract data from each publication according to these definitions. *Fig 1* shows typical waveforms for constant current pulsed stimulation and identifies selected characteristics, while Table 2. defines the parameters that were used to characterise the tSCS administered.

The root mean square (RMS) current is useful for estimating average electrical power and therefore the heat generating capacity of a waveform, $P_{avg} = i_{rms}^2 R$. For a square wave such, as at *Fig 1B*, the RMS current calculation simplifies to:

$$i_{rms} = i\sqrt{\frac{t1}{T}}$$

Or, for a typical symmetric biphasic waveform like that at *Fig 1A*, the calculation would be:

$$i_{rms} = i\sqrt{\frac{t1 + t2}{T}}$$

For the descriptions of other details of included studies, ranges are given with the mean ± standard deviation. Due to the heterogeneity in the methods used to evaluate the outcomes and the diverse experimental methodologies, a meta-analysis was not possible, and a descriptive qualitative review was conducted.

## Results

### Literature search and selection

Of the 3753 articles identified (Embase: 1960, Medline (Ovid): 1425, Web of Science:368), 2499 were taken to title and abstract screening after the duplicates were removed. After the removal of 2391 articles from title and abstract screening, 108 full texts were evaluated for

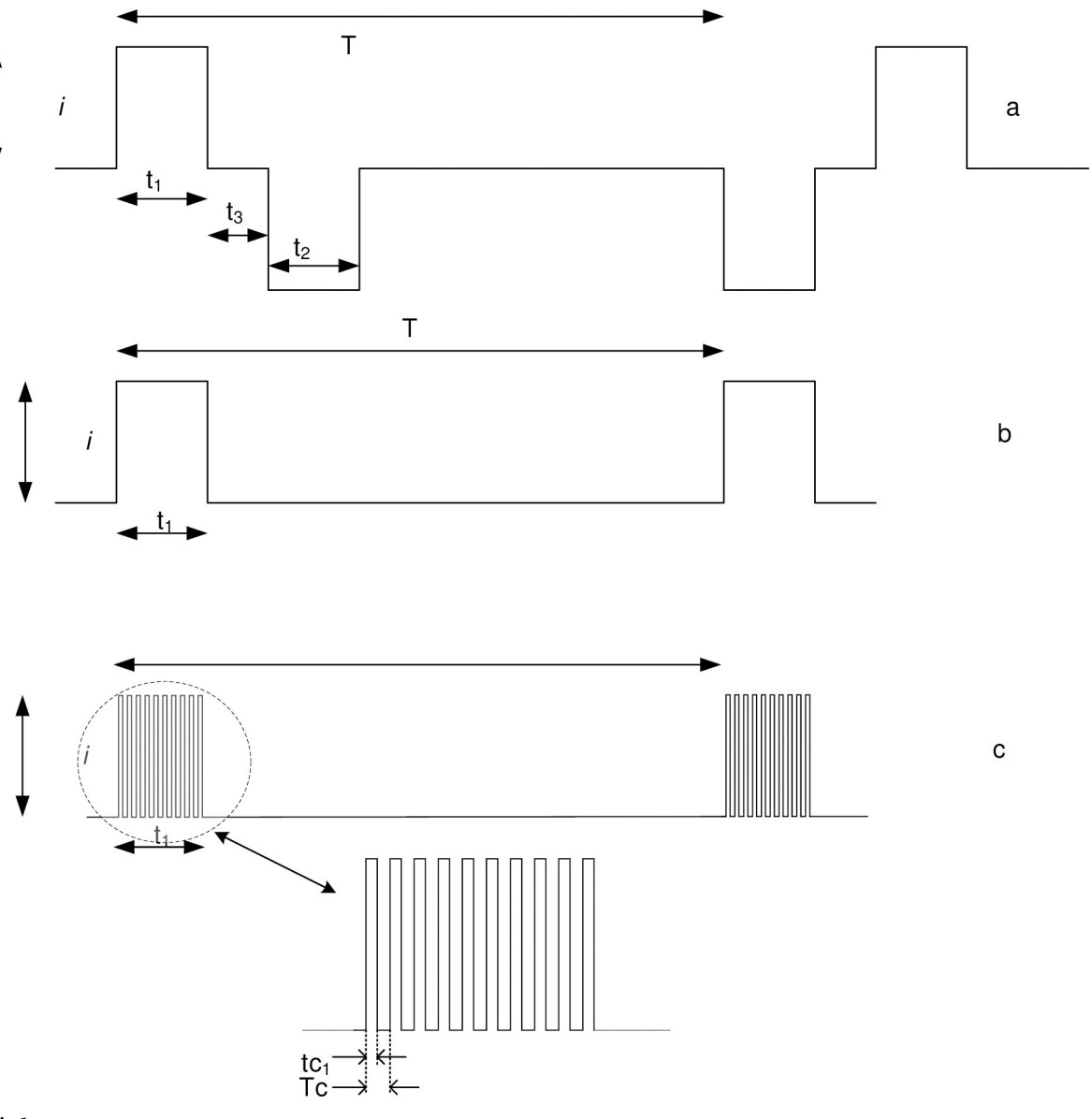

**Fig 1.**

eligibility. Finally, 25 articles that assessed the ability of tSCS to generate motor responses in individuals with SCI were included in this review (*Fig 2*).

## Study characteristics

Studies were categorized as neurophysiological assessments if their objective was to investigate the properties, mechanisms, or effects of tSCS on outcomes related to nervous system functioning (n = 9), whereas studies were labelled as therapeutic if they aimed to enhance motor rehabilitation and recovery in patients with SCI (n = 16). In therapeutic investigations, tSCS was commonly combined with simultaneous rehabilitative interventions such as physical therapy, treadmill training, body weight support and the use of exoskeletons or pharmacological agents (Table 3). Of the 25 included studies, 7 were case reports, 6 were case series, 3 were

**Table 2. Summary of stimulation parameters and a detailed description of how they are defined.** Definitions and characteristics of stimulation parameters.

| Parameter | Symbol | Unit | Description |
|---|---|---|---|
| Pulse interval | T | ms | The time interval between pulses of a sequence |
| Pulse frequency | f | Hz | The inverse of the pulse interval, f = 1/T, is the number of pulses per second |
| Phase duration | $t_1$ | ms | The duration of the leading phase |
| Pulse amplitude | i | mA | Current amplitude measured baseline to peak |
| Phase charge | $q_c$ | µC | Total charge in the phase |
| Pulse duration | p | ms | The sum of $t_1 + t_2 + t_3$ |
| Carrier frequency | $f_c$ | Hz | Frequency of a carrier waveform which is modulated by the stimulation waveform |
| Carrier-on-time | $tc_1$ | µs | Phase duration of carrier waveform |
| Carrier period | Tc | µs | Inverse of carrier frequency |
| Phase charge density | $q_d$ | µC/cm$^2$ | The phase charge per unit electrode area |
| Root mean square current | $i_{rms}$ | mA | $$i_{rms} = \sqrt{\frac{1}{T}\int_0^T i(t)^2 dt}$$ |
| Electrode current density | $j_e$ | mA/cm$^2$ | $j_e = i_{rms}/A$ |
| Electrode area (active) | A | cm$^2$ | The area of electrical contact at the skin. (assumed uniform current distribution within electrode) |

crossover trials, 7 were quasi-experimental studies (non-equivalent control group or nonrandomised intervention design), one was a sample cohort study, and one was a non-randomised control trial.

## Participant demographics

A total of 173 participants with SCI were recruited across the 25 studies to receive tSCS and their characteristics are described in Table 3. Further analysis only includes data from participants with SCI due to the purposes of this review. The sample sizes in included studies were generally modest (n = 7 ± 6). Neurophysiological investigations tended to have larger samples (n = 10 ± 5) than therapeutic investigations (n = 5 ± 5). A large range of ages, from 18 to 70 (mean 35 ± 13 years), injury classifications (from the level of C1—L2), and impairment levels (AIS A—D classifications), were represented across the included studies. Studies explored the effects of tSCS on different injury chronicity's, from one year to 43 years post-injury occurrence, however, no published studies investigated the use of tSCS at < 1-year post injury.

## Quality appraisal

The quality of included trials was evaluated using the D&B Checklist [58] and this tool deemed the overall evidence quality to be poor-to-fair, with results ranging from 3 to 19, out of a possible score of 32 (Tables 4 and 5). The mean score across all trials was 10.5 ± 4.9, with 11.4 ± 5.1 for neurophysiological and 10.0 ± 5.1 for therapeutic investigations. In particular, low scores were repeatedly awarded for external validity and selection bias, and the majority of studies were deemed insufficiently powered.

## Methodological evaluation

The methodologies of selected studies were reviewed to outline the common procedures for stimulation implementation and outcome evaluation. The stimulation variables selected

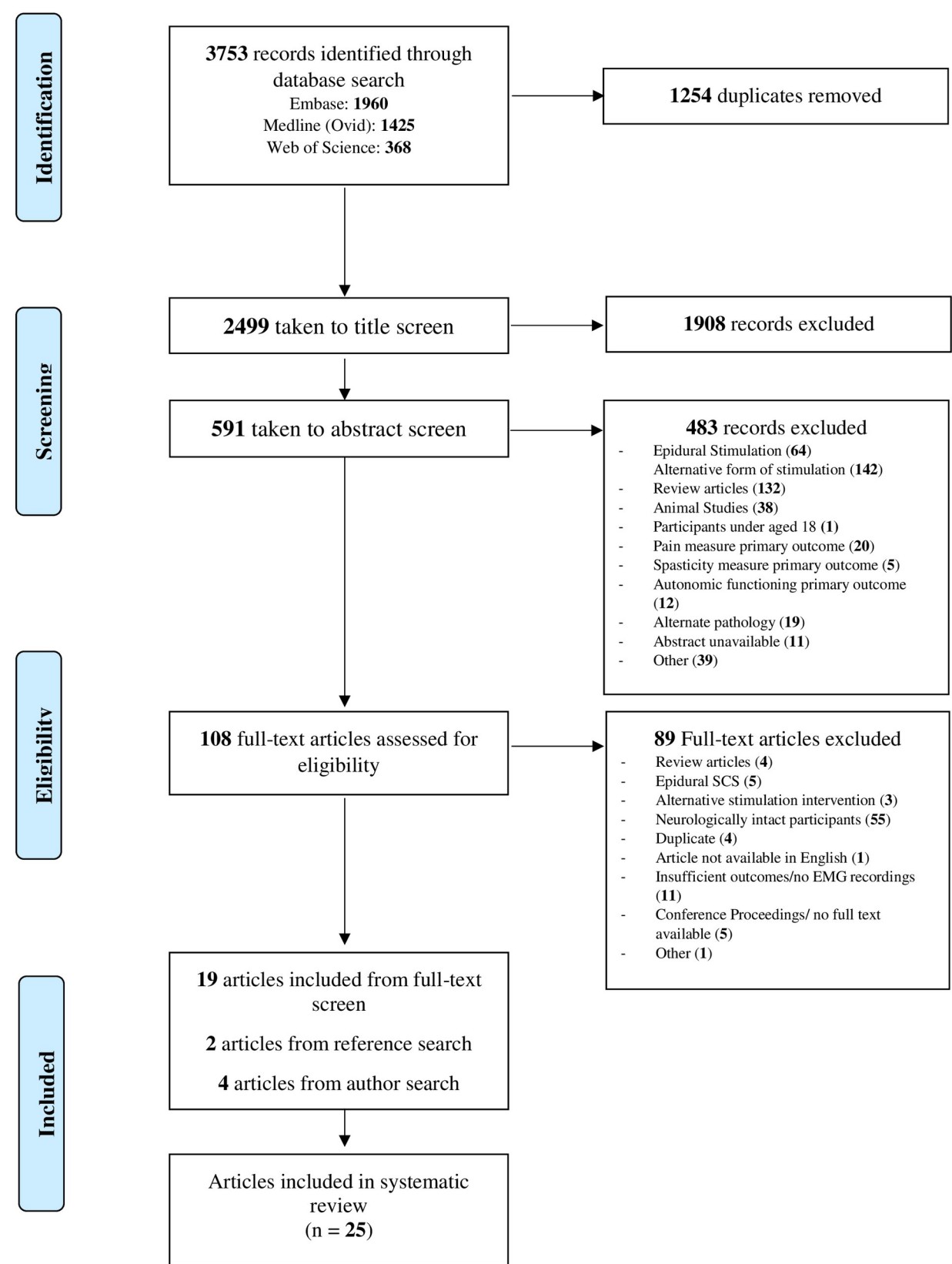

**Fig 2. PRISMA flow diagram of screening and selection processes.**

**Table 3. Details of study characteristics and the demographics of included participants.** Study characteristics and participant demographics.

| Study | Design | Simultaneous Interventions | Other Subjects (*n*) | Sample (*n*) | Age; Mean (SD) | Gender | AIS | Level | Years Since Injury; Mean (SD) |
|---|---|---|---|---|---|---|---|---|---|
| | | | | Participants with SCI Demographics | | | | | |
| Dy *et al.*, 2010 [14] | NP, QE | Treadmill and BWS | 9 NI | 9 | 32.6 (9.2) | M = 9 | A | C5-T7 | 6.4 (9.4) |
| Hofstoetter *et al.*, 2013 [6] | TP, CR | Treadmill | - | 1 | 29 | F = 1 | D | T9 | 11 |
| Gerasimenko *et al.*, 2015 [67] | TP, CS | Assisted movement, Buspirone | - | 5 | 31.4 (16.8) | M = 5 | B | C5-T4 | 3.2 (1.6) |
| Hofstoetter *et al.*, 2015 [18] | TP, CS | Treadmill | - | 3 | 32.7 (5.0) | M = 2, F = 1 | D | C5, T9 | 10.7(1.5) |
| Bedi, 2016 [68] | TP, CR | - | - | 1 | 25 | M = 1 | C | T12 | - |
| Emeliannikov *et al.*, 2016 [15] | NP, CS | Seated gait device, pharmacology | - | 10 | 39.1 (11.3) | M = 7, F = 3 | A-D | T5-L2 | 4.8 (4.2) |
| Minassian *et al*, 2016 [19] | TP, CS | RDGO, Treadmill and BWS | - | 4 | 39.5 (17.1) | M = 3, F = 1 | A | C8-T8 | 2.8 (1.4) |
| Gad *et al.*, 2017 [41] | TP, CR | Exoskeleton, Buspirone | - | 1 | 40 | M = 1 | A | T9 | 4 |
| Freyvert *et al.*, 2018 [44] | TP, CrT | Buspirone, hand grip exercises | - | 6 | 19.2 (1.3) | M = 4, F = 2 | B | C5-C8 | 2.4 (0.9) |
| Gad *et al.*, 2018 [43] | TP, CS | Hand grip exercises | - | 6 | 40.2 (16.6) | M = 5, F = 1 | B, C | C4-C8 | 8.0 (7.7) |
| Hofstoetter *et al.*, 2018 [16] | NP, QE | PT | 7 SCI eSCS | 10 | 39.7 (20.1) | M = 7, F = 3 | A, C, D | C4-T7 | 4.5 (2.8) |
| Inanici *et al.*, 2018 [36] | TP, CR | Activity-based PT | - | 1 | 62 | M | D | C3 | 2 |
| Rath *et al.*, 2018 [37] | TP, RCrT | - | - | 8 | 29.4 (7.7) | M = 7, F = 1 | A, C | C4-T9 | 7.3 (3.3) |
| Hofstoetter *et al.*, 2019 [17] | NP, QE | - | 10 NI | 10 | 40.1 (18) | M = 8, F = 2 | A, C, D | C4-T7 | 9.7 (12.5) |
| Murray and Knikou, 2019 [20] | NP, QE | - | 10 NI | 10 | 36.3 (11.2) | M = 7, F = 3 | A-D | C6-T12 | 8.8 (8.1) |
| Sayenko *et al.*, 2019 [42] | TP, RCrT | Stand Training | - | 15 | 31.2 (8.7) | M = 12, F = 3 | A-C | C4-T12 | 6 (3.2) |
| Al'Joboori *et al.*, 2020 [69] | Coh | BWS sit-to-stand training, standing exercises | 2 SCI Control | 5 | 36.8 (2.5) | M = 3, F = 2 | A, C, D | C6/7 – T10 | 3 (3.4) |
| Alam *et al.*, 2020 [40] | TP, CR | Stand Training and Treadmill | - | 1 | 48 | F = 1 | D | C7 | 21 |
| Atkinson *et al.*, 2020 [26] | NP, QE | - | 15 NI | 18 | 29.4 (7.3) | M = 16, F = 2 | A-D | C2-T6 | 4.6 (3.1) |
| Meyer *et al.*, 2020 [70] | TP, CS | Overground walking with BWS, ankle movement | - | 10 | 25.4 (12.4) | M = 9, F = 1 | D | C3-T10 | 11.6 (10.2) |
| Militskova *et al.*, 2020 [25] | NP, CR | PT, treadmill and BWS | - | 1 | 21 | F = 1 | A | T11 | 1 |
| Shapkova *et al.*, 2020 [38] | TP, NRCT | Exoskeleton | 16 SCI Control | 19 | 31.2 (8.6) | M = 15, F = 4 | A-C | C8-L2 | 4.6 (3.3) |
| Wu *et al.*, 2020 [23] | NP, QE | - | 14 NI, 4 ALS | 13 | 45.9 (13.7) | M = 10, F = 3 | B-D | C2-C8 | 10.8 (5.9) |
| Zhang *et al.*, 2020 [71] | TP, CR | Task specific hand training | - | 1 | 38 | M = 1 | A | C5 | 15 |
| Islam *et al.*, 2021 [72] | NP, QE | Treadmill walking and robotic gait orthosis BWS | 13 NI | 5 | 43.8 (11.4) | M = 9, F = 1 | B-D | C1- T11 | 13.4 (9.0) |

Abbreviations: AIS; ASIA impairment scale, ALS; amyotrophic lateral sclerosis, BWS; Body weight support, Coh; cohort study, CR; case report, CrT; crossover trial, CS; case series, eSCS; epidural spinal cord stimulation, F; female, M; male, NI; neurologically intact, NP; neurophysiological investigation, NRCT; non-randomised controlled trial, PT; physical therapy, QE; quasi-experimental study, RDGO; Robotic driven gait orthosis, RCrT; randomised crossover trial, SCI; spinal cord injury, SD; standard deviation, TP; therapeutic investigation.

**Table 4. Results from the quality appraisal of neurophysiological investigations using the Down's and Black Checklist.** Quality of Evidence Assessment–Neurophysiological Investigations.

| Study | Reporting | External Validity | Internal Validity (Bias) | Internal Validity (Selection Bias) | Power | Quality Score | Percentage | Evidence Category |
|---|---|---|---|---|---|---|---|---|
| Dy et al., 2010 [14] | 6 | 0 | 4 | 0 | 0 | 10 | 31% | Poor |
| Emeliannikov et al., 2016 [15] | 2 | 0 | 2 | 0 | 0 | 4 | 12% | Poor |
| Hofstoetter et al., 2018 [16] | 7 | 0 | 4 | 1 | 0 | 12 | 37% | Poor |
| Hofstoetter et al., 2019 [17] | 8 | 0 | 5 | 1 | 5 | 19 | 59% | Fair |
| Murray and Knikou, 2019 [20] | 7 | 0 | 3 | 0 | 1 | 11 | 34% | Poor |
| Atkinson et al., 2020 [26] | 6 | 0 | 4 | 0 | 0 | 10 | 31% | Poor |
| Militskova et al., 2020 [25] | 4 | 0 | 1 | 0 | 0 | 5 | 16% | Poor |
| Wu et al., 2020 [23] | 9 | 0 | 3 | 1 | 5 | 18 | 56% | Fair |
| Islam et al., 2021 [72] | 6 | 0 | 3 | 0 | 5 | 14 | 44% | Poor |
| | /11 | /3 | /7 | /6 | /5 | /32 | /100% | |

determine the electrical field generated and subsequent motor responses, and the outcomes used to evaluate these responses are essential in understanding the utility of the parameters selected and the overall effectiveness of tSCS. Factors such as safety and adverse events are also critical to a methodological review.

**Table 5. Results from the quality appraisal of therapeutic investigations using the Down's and Black Checklist.** Quality of Evidence Assessment–Therapeutic Investigations.

| Study | Reporting | External Validity | Internal Validity (Bias) | Internal Validity (Selection Bias) | Power | Quality Score | Percentage | Evidence Category |
|---|---|---|---|---|---|---|---|---|
| Hofstoetter et al., 2013 [6] | 1 | 0 | 2 | 0 | 0 | 3 | 9% | Poor |
| Gerasimenko et al., 2015 [67] | 4 | 0 | 2 | 0 | 0 | 6 | 19% | Poor |
| Hofstoetter et al., 2015 [18] | 2 | 0 | 2 | 0 | 0 | 4 | 12% | Poor |
| Bedi, 2016 [68] | 4 | 0 | 3 | 0 | 0 | 7 | 22% | Poor |
| Minassian et al., 2016 [19] | 5 | 0 | 4 | 0 | 0 | 9 | 28% | Poor |
| Gad et al., 2017 [41] | 3 | 0 | 0 | 0 | 0 | 3 | 9% | Poor |
| Freyvert et al., 2018 [44] | 7 | 0 | 5 | 2 | 0 | 14 | 44% | Poor |
| Gad et al., 2018 [43] | 9 | 0 | 2 | 0 | 0 | 11 | 34% | Poor |
| Inanici et al., 2018 [36] | 5 | 0 | 3 | 1 | 0 | 9 | 28% | Poor |
| Rath et al., 2018 [37] | 7 | 0 | 4 | 1 | 0 | 12 | 37% | Poor |
| Sayenko et al., 2019 [42] | 6 | 0 | 3 | 4 | 4 | 17 | 53% | Fair |
| Al'Joboori et al., 2020 [69] | 8 | 1 | 4 | 2 | 0 | 15 | 47% | Poor |
| Alam et al., 2020 [40] | 3 | 0 | 2 | 4 | 0 | 9 | 28% | Poor |
| Meyer et al., 2020 [70] | 8 | 0 | 4 | 3 | 1 | 16 | 50% | Poor |
| Shapkova et al., 2020 [38] | 8 | 1 | 5 | 0 | 3 | 17 | 53% | Fair |
| Zhang et al., 2020 [71] | 5 | 0 | 3 | 0 | 0 | 8 | 25% | Poor |
| Total Maximum Score | /11 | /3 | /7 | /6 | /5 | /32 | /100% | |

## Electrode configurations

The electrode configurations with regard to position and location varied substantially across experiments (Tables 6 and 7). The cathode was positioned dorsally over the vertebral column in the majority of studies, otherwise electrodes were used that alternated polarity within a biphasic pulse [16–18, 23, 40, 69, 70]. Some studies specified a paravertebral dorsal electrode orientation [16–19, 68], whereas most others placed the electrode in the midline over the vertebral column [14, 20, 23, 25, 26, 36, 37, 40–44, 67, 69–72]. Many studies targeted a single site, however, 8 out of 14 therapeutic investigations favoured the stimulation of multiple sites simultaneously [36, 37, 40–43, 67, 71].

The most common vertebral level stimulated for targeting lower limb motor activity was T11-T12 and/or L1-L2. Two studies placed electrodes within the range of T9-L2, but adjusted the exact positions based on evoked EMG motor responses [25, 26]. An additional secondary stimulating electrode was also placed on the coccygeal bone during two experiments [41, 67]. For the upper limb responses, the cathode site varied substantially across the four studies and was placed on C5 [44], T2-T4 [23], or C3-C4 simultaneously with C6-C7 or C7-T1 [36, 43, 71].

The anode location selected for experiments targeting lower limb motor responses varied between the anterior superior iliac spine (ASIS) and iliac crests (n = 6) or para-umbilically over the anterior abdomen (n = 8), with two studies recording the use of both locations depending on patient comfort [20, 72]. Conversely, one study described placing both the anode and the cathode in the midline over the vertebral column [69]. In studies of upper limb responses, the iliac crests or ASIS were chosen by four investigations [36, 43, 44, 71] with only Wu *et al.*, [23] placing the anode on the anterior neck.

## Electrical dosage

Clear differences in dosage arose between neurophysiological and therapeutic investigations. In neurophysiological investigations, tSCS was typically delivered using isolated single or paired pulses with long refractory periods allowing a return of resting membrane potential and frequencies, when outlined, were typically low. A notable exception to this was a recent investigation by Islam et al. [72] who administered high frequency pulse trains at 333Hz, the rationale for which was unclear. Delivered current in neurophysiological investigations ranged between 24.7 [14] to 248 ± 87.06 mA [72] with only two studies exceeding a maximum of 100mA [20, 72]. A variety of criteria were used to determine the stimulation intensities, for example, the point at which threshold responses were observed in some [20, 23] or all muscles [14, 16, 17], maximum tolerance [25], response magnitude plateau [26], equivalent amplitude to H-Reflex soleus response [17, 20, 72], or the lowest amplitude that completely supressed the second stimulus of a pair [15]. The majority of neurophysiological experiments reported using a square or rectangular monophasic current waveform with 1ms pulse width, with just two studies using biphasic pulses [16, 17], and one which trialled both [23].

In contrast, therapeutic investigations typically reported the application of continuous pulse trains, with a burst frequency of 5–30 Hz and an intra-pulse carrier frequency of 2.5–10 kHz [36, 37, 42, 43, 67, 68]. The use of this intra-pulse carrier frequency is poorly justified and appears to be for analgesic purposes, although no evaluation of this could be identified. Other therapeutic experiments selected simplified phase characteristics with either biphasic or mono-phasic rectangular-waves with a frequency ranging from 1–90 Hz, with 20–30 Hz the most commonly occurring selection [6, 18, 19, 38, 40, 44, 69, 71]. The duration of therapeutic stimulation varied from bouts of <5 mins [37, 40, 67] to > 45 mins [36, 38, 68] and was generally paired with concomitant rehabilitative activities. Recorded current ranges in therapeutic experiments reached a maximum of 180mA in the thoracolumbar region [67] and 210mA in

**Table 6. Parameters selected by neurophysiological assessments investigating the effects of tSCS on spinal cord functioning in individuals with SCI.** Stimulation parameters selected by studies carrying out neurophysiological assessments into the properties of spinal cord stimulation with SCI participants.

| Study | Patient position | ELECTRODES | | | STIMULATION PROTOCOL | | | ELECTRICAL CHARACTERISTICS | |
| | | Size/shape [Area] | Polarity | Location | Description | Frequency | Intensity | Phase Charge (μC) | Max Phase Charge Density (μC/cm$^2$) |
|---|---|---|---|---|---|---|---|---|---|
| *Lower limb responses* | | | | | | | | | |
| Dy *et al.*, 2010 [14] | Lying prone, BWS standing, BWS stepping | ø 2.5 cm [4.9 cm$^2$] | Cathode | T11-T12 | Single, t$_1$ = 1ms, mono square wave pulses | 1) Prone/ Standing: 0.5 Hz | 24.7 - 83mA | 25–83 | 16.9 |
| | | Pair 5.0 x 10.2 cm | Anodes | Iliac crests | | 2) Stepping: 0.25–0.33 Hz | Set to where consistent responses observed in all measured muscles in standing | | |
| Emeliannikov *et al.*, 2016 [15] | Seated | - | - | T11-T12 | t$_1$ = 1ms paired pulses (50ms inter-pulse interval) | 0.3 Hz for H-Reflex | 30–80 mA | 30–80 | - |
| | | - | - | - | | | Lowest amplitude to completely supress the second stimulus of a pair | | |
| Hofstoetter *et al.*, 2018 [16] | Lying supine | Pair ø 5 cm [2 x 19.6 cm$^2$] | Alternating (anode first pulse, cathode second) | T11-T12 paravertebrally | Charge balanced, symmetric biph rectangular t$_1$ = 1ms | - | 32–86 mA Adjusted to reach target threshold >100uV in all muscle groups studied | 32–86 | 2.2 |
| | | 8 x 13 cm | Alternating | Para-umbilically lower abdomen | | | | | |
| Hofstoetter *et al.*, 2019 [17] | Lying supine | 5 x 9 cm [45 cm$^2$] | Alternating (anode first pulse, cathode second) | T11—T12 paravertebrally | Charge balanced, symmetric biph rectangular t$_1$ = 1ms | - | Adjusted to elicit control-PRM reflexes in the right soleus with amplitudes that best matched the control-H reflexes and to elicit PRM reflexes in other muscles studied | - | - |
| | | Pair 8 x 13 cm | Alternating | Lower abdomen | | | | | |
| Murray and Knikou, 2019 [20] | Lying supine | 10.2 x 5.1 cm [52 cm$^2$] | Cathode | T10—L1/L2 | 1) Intervention: alternating suprathreshold and subthreshold stimulation (60 mins/session), mono square wave t$_1$ = 1ms | 1) 0.2Hz | Selected based on threshold to produce right soleus evoked potential (96.9 ± 24 mA). Treatment sessions ranged from 0.4–4.3 times this resting threshold | 97 | 1.9 |
| | | Connected pair 10.2 x 5.1 cm$^2$ | Anode | Para-umbilically/ iliac crests | 2) Assessment: mono square wave t$_1$ = 1ms | 2) 0.1, 0.125, 0.2, 0.33, 1.0 Hz | From below motor threshold until plateau reached | 417 | 8.0 |
| Atkinson *et al.*, 2020 [26] | Lying supine | ø 1.8 cm [2.6 cm$^2$] | Cathode | Midline T9-T10 (n = 1), T10-T11 (n = 7), T11-T12 (n = 6), T12-L1 (n = 1) | Single, mono square wave pulses, t$_1$ = 1ms | - | 0-100mA or until response magnitude plateaued | 0 to 100 | 39.3 |
| | | Pair 5 x 9 cm | Anode | Anterior superior iliac spines | | | | | |

(*Continued*)

**Table 6.** (Continued)

| Study | Patient position | Size/shape [Area] | Polarity | Location | Description | Frequency | Intensity | Phase Charge (μC) | Max Phase Charge Density (μC/cm$^2$) |
|---|---|---|---|---|---|---|---|---|---|
| | | ELECTRODES | | | STIMULATION PROTOCOL | | | ELECTRICAL CHARACTERISTICS | |
| Militskova et al., 2020 [25] | Lying supine, BWS standing | ø 2.5 cm [4.9 cm$^2$] | Stimulating | Midline T9-T10, T10-T11, T11-T12, T12-L1, and L1-L2 | Mono rectangular pulses t1 = 1ms | - | 30-100mA or maximum tolerated | 30 to 100 | 20.4 |
| | | Pair 4 x 2 cm | Reference | Lower abdomen | | | | | |
| Islam et al., 2021 [72] | Lying supine, BWS stepping | 10.2 × 5.1 cm [52 cm$^2$] | Cathode | Longitudinally between T10-L1 vertebrae | 1) Single mono pulses t1 = 1ms | | 248 ± 87.06 mA for single pulses, | 248 ± 87.06 | 4.8 |
| | | pair 10.2 × 5.1 cm | Anode | Iliac crests or either side of abdominal muscles | 2) Pulse train of 12 1 ms pulses with a total duration of 33 ms randomly across the step cycle | 333.3 Hz | 57 to 160 mA for pulse trains Intensity set for Sol TEPs to be equivalent to the Sol H-reflex | 57 to 160 | 18.8 to 53 |
| **Upper limb responses** | | | | | | | | | |
| Wu et al., 2020 [23] | Seated | 5 x 10cm [50 cm$^2$] | 1) Alternating polarity | 4cm caudal to C7 (T2-T4) | 1) Anode posterior 2ms, t1 = 1ms biph, | 0.2Hz | 80–175% of RMT, (RMT = 5.5–51 ma) | 89 μc mono | 1.8 |
| | | | 2) Cathode posterior for majority | | 2) Cathode posterior 2ms, t1 = 1ms biph, | | | 89 μc biph | 1.8 |
| | | 5 x 10cm | 1) Alternating polarity | 1-2cm above sternal notch (C4-C5 levels anteriorly) | 3) Cathode posterior t1 = 0.5ms biph, | | Threshold calculated as enough to elicit > 50uv in 5/10 reps | | |
| | | | 2) Anode anterior for majority | | 4) Cathode posterior t1 = 1ms mono | | | | |

**Abbreviations:** biph; biphasic, mono; monophasic, PRM; posterior root-muscle, RMT; resting motor threshold, Sol; soleus, TEP; transpinal evoked potential. Where more than one test protocol existed within a given publication, the protocols were detailed using numerical listing: 1) 2) 3) etc. Phase charge density is given in terms of upper limits.

the cervical region [43]. Intensity criteria was not always explicitly specified. Some studies note that it was based off sufficient levels to reach desired muscle responses [43, 67], perceived sensory thresholds [68] or the amplitude at which reflex threshold was reached [19]. The pulse width was between 0.5 to 1 ms per phase with rectangular waveforms and the majority applied monophasic pulses, with the exception of several studies that selected biphasic pulses [6, 18, 40, 69, 70] and two studies that used both [36, 43]. Two studies used voltage pulses [6, 18] and the resulting current amplitude was not available.

## Electrical characteristics

The variances in electrode sizes and configurations along with differences in dosage parameters such as amplitude, frequency and pulse duration make it difficult to compare the electrical characteristics of stimulation across studies. We have therefore attempted to calculate common characteristics that were gleaned from the available data. The pulse charge was reasonably

**Table 7. Parameters selected for therapeutic stimulation investigating the effects of tSCS on motor rehabilitation.** Stimulation parameters selected by therapeutic studies investigating the effects of transcutaneous spinal cord stimulation for motor recovery.

| Study | Position/ Action | ELECTRODES | | | STIMULATION PROTOCOL | | | | ELECTRICAL CHARACTERISTICS | | |
| | | Size/shape [Area] | Polarity | Location | Description | Frequency | Intensity | Duration | Pulse Charge (μC) | Current RMS (mA) | Current Density (mA/ cm$^2$) |
|---|---|---|---|---|---|---|---|---|---|---|---|
| Hofstoetter et al., 2013 [6] | Upright/ stepping on treadmill | Pair ø 5cm [2 x 19.6 cm$^2$] | - | Sgl: T11/T12 | Sub-motor threshold, charge balanced, symmetric, biph rectangular pulses of $t_1$ = 1ms | 30Hz | 18V | - | - | - | - |
| | | Pair 8 x 13cm | - | Lower anterior abdomen | | | | | | | |
| Gerasimenko et al., 2015 [67] | Side-lying/ gravity-neutral stepping | ø 2.5cm [4.9 cm$^2$] | Cathode | Mult: T11-T12 and coccyx 1 (Co1) | Monopolar rectangular stimuli, $t_1$ = 1ms | T11: 30Hz (+10kHz cf) | 80-180mA, Stepping motor threshold | 3 x 3 mins (T11, Co1, both) | 40–90 | 9.8–22 | 2.0–4.5 |
| | | Pair 5 x 10.2cm$^2$ | Anode | Iliac crests | | Co1: 5Hz (+10kHz cf) | | | 40–90 | 4.0–9.0 | 0.8–1.8 |
| Hofstoetter et al., 2015 [18] | Standing/ treadmill stepping | Pair ø 5cm [2 x 19.5 cm$^2$] | Alternating (anode 1st phase, cathode 2nd) | Sgl: T11/T12 paravertebrally | Charge-balanced, symmetric, biph rectangular $t_1$ = 1ms | 30Hz | 18–27 V, 86% of reflex threshold (P1), 71% (P2), 80% (P3). | - | - | - | - |
| | | Pair 8 x 13cm | Alternating | Para-umbilically | | | | | | | |
| Bedi, 2016 [68] | Voluntary and passive movement | Pair 4.5 x 9 cm [2 x 41 cm$^2$] | - | Sgl: T10-L1 paravertebrally - | - | 30, 50, 70, 90 Hz (+2.5 kHz cf) | Raised to sensory threshold | 45 mins per frequency | - | - | - |
| Minassian et al., 2016 [19] | Supine, Standing/ assisted treadmill stepping | Pair ø 5 cm [2 x 19.6 cm$^2$] | Cathode | Sgl: T11- T12, paravertebrally, 1cm apart | Rectangular mono $t_1$ = 1ms | 30Hz | P1: 140mA | 10 gait cycles | P1: 140 | 24.25 | 0.62 |
| | | | | | | | P2: 100 mA | | P2: 100 | 17.32 | 0.44 |
| | | | | | | | P3:170 mA | | P3: 170 | 29.44 | 0.75 |
| | | | | | | | P4: 125 mA | | P4: 125 | 21.65 | 0.55 |
| | | Pair 8 x 13cm | Anode | Abdomen | | | Increments of 5ma until reflex threshold | | | | |
| Gad et al., 2017 [41] | Standing, supine/ exoskeleton stepping, voluntary movement | ø 2.5cm [4.9 cm$^2$] | Cathode | Mult: T11-T12 and coccyx 1 (Co1) | - | T11: 30H | - | 3 x 20 mins/ session | - | - | - |
| | | | | | | Co1: 5Hz | | | | | |
| | | 5 x 10.2cm$^2$ | Anode | Iliac crests | | Tc = 100μs cf | | | | | |
| Sayenko et al., 2019 [42] | Standing/ standing balance exercises, sit-to-stand | ø 3.2cm diameter [8 cm$^2$] | Cathode | Mult + sgl: | Mono $t_1$ = 1ms pulses | 1) T11/L1: 5, 15, 25 Hz Tc = 100μs cf | 1) T11/L1: Up to 150 mA | - | 75 | 16.7 (@ 25 Hz) | 2.1 |
| | | | | 1) T11 and/or L1 | | | | | | | |
| | | | | 2) L1 | | | | | | | |
| | | Pair 7.5 x 13 cm | Anode | Iliac crests | | 2) L1: 15Hz | 2) L1: Up to 100 mA | | 75 | 8.7 (@15 Hz) | 1.1 |
| Al'Joboori et al., 2020 [69] | Standing/ sit-to-stand training and standing exercises | 5×5 cm [25 cm$^2$] | Cathode | T10/11 | 2) Biph $t_1$ = 1ms | 2) 30 Hz | Below motor threshold | 60 mins per session | 110 | 26.9 | 1.1 |
| | | 5×5 cm | Anode | T12/L1 | | | 40–110 mA | | | | |

(*Continued*)

**Table 7.** (Continued)

| Study | Position/ Action | ELECTRODES | | | STIMULATION PROTOCOL | | | | ELECTRICAL CHARACTERISTICS | | |
|---|---|---|---|---|---|---|---|---|---|---|---|
| | | Size/shape [Area] | Polarity | Location | Description | Frequency | Intensity | Duration | Pulse Charge ($\mu$C) | Current RMS (mA) | Current Density (mA/ cm$^2$) |
| Alam *et al.*, 2020 [40] | Standing, sitting, supine/ standing, stepping and voluntary movement training | Pair ø 3.2cm [2 x 8 cm$^2$] | Alternating | Mult: T11 and L1 | 1) Biph with Tc$_1$ = 50µsec T$_1$ = 100 µs | 1) 20 Hz /100 µs | 1) T11: 105ma, L1: 100 mA | 3 x ~10 mins and 3 x 2–3 mins/ session | 5.3 | 3.8 | 0.3 |
| | | Pair 6x9 cm | | Iliac crests | 2) Tonic biph stimulation with Tc$_1$ = 50µs T$_1$ = 100 µs | 2) 30 Hz /100 µs | 2) T11: 95ma, ll: 90 mA | | 4.8 | 4.7 | 0.3 |
| | | | | | 3) Tonic biph stimulation with Tc$_1$ = 50µs t$_1$ = 1ms | 3) 20–30 Hz/1ms | 3) T11: 20-120ma, L1: 20–120 mA | | 60 | 14.7 (@30Hz) | 0.9 |
| Meyer *et al.*, 2020 [70] | Supine, standing/ stepping and voluntary movement training | 5 × 9 cm [45 cm$^2$] | Alternating (anode for 1$^{st}$ phase, cathode 2$^{nd}$) | T11-t12 | Charge-balanced, symmetric, biph rectangular t$_1$ = 1ms | 15 hz, 30 Hz, and 50 Hz | 0.8–1.0 x PRM-reflex threshold, 1) Ankle movements 26.4 ± 17.3 mA | 1) Ankle control and 2) walk test: max 5 min | 26.4 | 4.6 (@15 Hz) | 0.1 |
| | | | | | | | | | | 8.4 (@50 Hz) | 0.2 |
| | | pair 7.5 × 13 cm | Alternating | Abdomen, paraumbilically | | | 2) Walk tests: 34.8 ± 13.1 ma | 3) Spinal reflex assessment: max 15 min | 34.8 | 6.0 (@15 Hz) | 0.13 |
| | | | | | | | 3) Spinal reflex: 28.8 ± 14.7 mA | | | 11 (@50 Hz) | 0.24 |
| Shapkova *et al.*, 2020 [38] | supine, standing/ exoskeleton walk training | Pair 3 x 4 cm [2 x 12 cm$^2$] | Cathode | Sgl: T12 vertebrae | T$_1$ = 0.5ms mono square wave | G1: 1 Hz, G2: 3 Hz, G3: 67 Hz | 1.3–1.4 x motor threshold | ~41–53 mins/ session | 50 | G1: 2.2 | 0.1 |
| | | | | | | | | | | G2: 3.9 | 0.2 |
| | | | | | | | | | | G3: 618.3 | 0.8 |
| | | Pair 3 x 4 cm | Anode | Central abdomen | | | 5–100 mA | | | | |
| *Upper limb responses* | | | | | | | | | | | |
| Freyvert *et al.*, 2018 [44] | Voluntary hand contractions | - | Cathode | Sgl: C5 | | 5-30Hz for 15–30 mins | 20–100 mA | 15–30 mins/ session | - | - | - |
| | | | Grounding | ASIS | | | | | | | |
| Gad *et al.*, 2018 [43] | Voluntary hand contractions | ø 2cm diameter [3.1 cm$^2$] | Cathode | Mult: C3-C4 and C6-C7 | 1) Pulsed monot$_1$ = 1ms | 1) 1Hz | 1) 10–200 mA | - | 100 | 4.5 (1 Hz) | 1.4 |
| | | Pair 5.0 x 10cm$^2$, rectangular | Anode | Iliac crests | 2) Continuous biph or mono t$_1$ = 1ms | 2) 30 Hz + (Tc = 100µs *cf*) | 2) 70–210 mA | | | 25.7 (@30Hz, bi) | 8.2 |
| | | | | | | | | | | 36.4 (@30Hz mono) | 11.6 |

(*Continued*)

**Table 7.** (Continued)

| Study | Position/ Action | Size/shape [Area] | Polarity | Location | Description | Frequency | Intensity | Duration | Pulse Charge (μC) | Current RMS (mA) | Current Density (mA/ cm²) |
|---|---|---|---|---|---|---|---|---|---|---|---|
| | | ELECTRODES | | | STIMULATION PROTOCOL | | | | ELECTRICAL CHARACTERISTICS | | |
| Inanici *et al.*, 2018 [36] | Upper limb activity-based PT | Pair x ø 2.5cm [2 x 8 cm²] | Cathode | Mult: midline C3-C4 and C6-C7 | 1) Continuous 60 ± 20 mins biph $t_1$ = 1 ms | 1) 30 Hz + (Tc = 100μs cf) | 1) 80–120 mA | 1) 60 ± 20 mins/ session | 60 | 14.7 (30Hz) | 1.5 |
| | | Pair x 5 x 10cm | Anodes | Iliac crests | 2) Pulsed mono rectangular $t_1$ = 1ms bursts | 2) 1Hz | 2) 10–120 mA at 10 mA intervals | 2) Pulsed | 60 | 2.7 (1Hz) | 0.3 |
| Zhang *et al.*, 2020 [71] | Seated/ Upper limb activity-based PT | Pair x round electrodes | Cathode | Mult: midline C3-C4, C7-T1 | Pulsed mono rectangular $t_1$ = 1ms bursts | 30 Hz + (Tc = 100μs cf) | 0–80 mA at 5 mA increments | Pulsed assessment | 40 | 9.8 @ 30Hz | 0.6 |
| | | 8 x 13cm | Anode | Iliac crests | | | C3-4: 50 mA | 60 mins/ session | 25 | 6.12 @ 30Hz | 0.4 |
| | | | | | | | C7-T1: 15 mA, | | | | |
| | | | | | | | Regularly adjusted | | | | |

*Trunk responses*

| Study | Position/ Action | Size/shape [Area] | Polarity | Location | Description | Frequency | Intensity | Duration | Pulse Charge (μC) | Current RMS (mA) | Current Density (mA/ cm²) |
|---|---|---|---|---|---|---|---|---|---|---|---|
| Rath *et al.*, 2018 [37] | Sitting/ seated balance tasks | 2 x ø 3.2cm [2 x 8 cm²] | Cathode | Mult: T11 and L1 | Mono, rectangular 1ms pulses | T11: 30Hz (Tc = 100μs cf) | 1) 10–150 mA to detect motor threshold | 3–4 x ~1–2 mins/ session | 50 | 12.25 (T11) | 1.52 |
| | | Pair 7.5 x 13cm | Anode | Iliac crests | | L1: 15Hz (Tc = 100μs cf) | 2) constant sub-threshold | | 40 | 6.93 (L1) | 0.86 |
| | | | | | | | T11: 25–100 mA | | | | |
| | | | | | | | L1: 5–80 mA | | | | |

Abbreviations: ASIS; Anterior Superior iliac Spine, biph; biphasic, cf; carrier frequency, Co1; coccyx 1, G; group, mono; monophasic, mult; multiple stimulation levels, P; participant, PT; physical therapy, sgl; single stimulation level.

Where more than one test protocol existed within a given publication, the protocols were detailed using numerical listing: 1) 2) 3) etc.

Electrical characteristics such as current RMS and current density include the upper limits.

consistent in the neurophysiological investigations, in the range 30 to 100 μC, although two studies did exceed this value [20, 72]. The resulting maximal charge density at the spinal electrode varied enormously between studies because of the range of electrode sizes used, 1.8 to 53 μC/cm².

The therapeutic investigations used sustained trains of pulses and the resulting current and current density was compared. Root-mean-square current was in the range 2.2 to 36.4 mA, with most studies below 20 mA. Once again, the variation in electrode area led to a wide range of current densities between studies, 0.1 to 4.5 mA/cm², with one study exceeding this reaching 11.6 mA/cm² due to a high current combined with a small electrode area [43].

## Participant positioning

The majority of neurophysiological investigations were conducted with subject in either supine [16, 17, 20, 26] or seated [15, 23] positions. A small number of studies were carried out with participants standing [14, 72] and prone [14]. One investigation compared a number of varying positions to investigate positional effects [25]. In contrast, the majority of therapeutic interventions targeting lower limb responses were conducted in an upright standing position [6, 18, 19, 38, 40–42, 69] and/or while supine [19, 38, 40, 41]. Stimulation targeting trunk control and sitting balance was carried out in a seated position [37] and seated positioning was only outlined in one study of upper limb functioning [71].

## The reflex nature of tSCS responses

Only four therapeutic investigations [18, 19, 69, 70] assessed the nature of motor responses generated by tSCS, while in contrast, all neurophysiological investigations recorded the reflex origin of evoked responses. Primarily the transynaptic modulation of responses was demonstrated using the paired pulse paradigm in which two pulses were delivered with a short conditioning-test interval (CTI) to demonstrate PAD of the second response. Interstimulus intervals between 30-50ms were generally selected to demonstrate PAD [15, 16, 23, 25, 69, 70], with a loss of amplitude attenuation of the second pulse occurring at intervals greater than 100ms [17, 19]. Other than the paired pulse paradigm, response latencies were also used to indirectly evaluate stimulation of dorsal afferents to trigger a reflex response [23], along with the use of vibration to demonstrate pre-synaptic inhibition of motor responses [14].

## Outcome measurement

There were a large variety of outcome measures employed by therapeutic investigations to evaluate motor performance, with 30 different measures used across 16 investigations (Table 8). A total of 10 studies measured joint kinematics, 7 studies assessed functional outcomes and 5 studies each assessed gait parameters and force production. Only two therapeutic investigations evaluated the effects of tSCS on subjective quality of life outcomes [36, 69]. Apart from the recording of EMG data, the most frequently employed objective outcomes in therapeutic studies were an evaluation of AIS scoring (n = 6), goniometer data of joint angles (n = 5), and centre of pressure/foot loading data (n = 5).

Neurophysiological investigations focused primarily on objectively evaluating the amplitude of EMG responses evoked from tSCS, although some studies additionally looked at the conditioning effects of tSCS on spinal excitability as measured by H-reflex and M-wave amplitude [15, 17, 20, 26, 72]. Temporal/phasic modulation of responses evoked by tSCS during gait were also assessed by two studies [14, 72].

## Surface electromyography

Surface EMG recordings from 28 different muscle locations on the lower limb (n = 11), upper limb (n = 13), and trunk (n = 4) were described. An overview of the recording, processing and presentation of EMG signals are presented in Tables 9 and 10 for neurophysiological and therapeutic studies, respectively. Only 8 studies provided adequate details of the preparation including skin preparation, electrode type, shape, composition, and inter-electrode distance [6, 14, 16–19, 68, 72], each experiment recorded the use of round silver- silver chloride electrodes with an interelectrode distance of 1.7, 2 or 3 cm. Sampling frequencies ranged from 1,000 to 10,000 Hz.

**Table 8. Outcomes measures employed by therapeutic studies investigating the outcomes of tSCS on motor rehabilitation in individuals with SCI.** Outcomes selected in the included studies evaluating the effects of therapeutic transcutaneous spinal cord stimulation in spinal cord injured individuals.

| Study | Force | Kinematics | Gait | Function | Other |
|---|---|---|---|---|---|
| Hofstoetter et al., 2013 [6] | - | joint angles (goniometer) | Stride length, cycle duration (pressure switches) | - | - |
| Gerasimenko et al., 2015 [67] | - | joint angles (goniometer) | - | - | - |
| Hofstoetter et al., 2015 [18] | - | joint angles (goniometer) | Swing/stance phase duration, cycle duration (foot sensor) | - | - |
| Bedi, 2016 [68] | - | - | - | - | - |
| Minassian et al., 2016 [19] | - | - | - | - | - |
| Gad et al., 2017 [41] | - | joint angles (goniometer and EKSO position sensors) | cycle duration (EKSO device) | - | Self-scoring: muscle tone, sensation, perspiration, coordination, level of robotic assistance, mean HR/BP during training |
| Freyvert et al., 2018 [44] | Handgrip force measurement | - | - | UEMS (AIS), ARAT | Spasticity (MAS) |
| Gad et al., 2018 [43] | Handgrip force measurement (transducer) | - | - | Motor and sensory scores (AIS) | Self-report QoL |
| Inanici et al., 2018 [36] | Pinch strength (pinch gauge) | - | - | AIS scoring, GRASSP | QoL questionnaires (WHO Quality of Life—BREF, SF-Qualiveen, SCIM III) |
| Rath et al., 2018 [37] | - | Video and 3D kinematic recordings (Xbox One Kinect), centre of pressure (force plate system) | - | - | - |
| Sayenko et al., 2019 [42] | Knee assistance (force sensing resistor) | Centre of pressure (force plate) | - | - | Qualitative level of assistance, time spent standing |
| Al'Joboori et al. 2020 [69] | - | Ankle and knee joint ROM (electro-goniometers), STS leg loading (force plates) | | Motor and Sensory scores (AIS) | BMCA, SF-36 health survey, SCIM III |
| Alam et al., 2020 [40] | - | joint angles and body positions (integrated motion capture system), Sit-to-stand transitions (force plate) | - | AIS scoring | - |
| Meyer et al., 2020 [70] | | Ankle ROM and gait kinematics (motion capture system) | gait speed | | |
| Shapkova et al., 2020 [38] | - | Joint angles and body position (ExoAtlet Global exoskeleton), foot loading (force plates and F-Scan sensors) | Hauser Ambulation Index, maximum nonstop walk duration (ExoAtlet Global exoskeleton), Asymmetry Index (ASI) | AIS scoring | Spasticity (MAS), spinal excitability (H-Reflex amplitude) |
| Zhang et al., 2020 [71] | Handgrip force, lateral pinch force, elbow flexion torque (dynamometer) | - | - | GRASSP, NRS | spinal motor evoked potentials |

**Abbreviations:** AIS; ASIA Impairment Scale, ARAT; Action Research Arm Test, ASI; asymmetry index, BMCA; Brain Motor Control Assessment, EKSO; Ekso Bionics, EMG; electromyography, GRASSP; Graded and Redefined Assessment of Strength, Sensibility and Prehension, MAS; Modified Ashworth Scale, NRS; Neuromuscular Recovery Scale, QoL; quality of life, SCIM III; Spinal Cord Independence Measure Version III, SF; short form, STS; sit-to-stand, UEMS; upper extremity motor score, WHO; World Health Organisation.

Several studies explicitly reported filters for stimulus artefact such as bandpass [67], Butterworth [6, 14, 41] or linear adaptive filters [37, 42], whereas others attempted to quantify stimulation artifact by placing electrodes on alternative trunk muscles that were not directly

**Table 9. A summary of evoked surface EMG data collection, recording and signal processing.** EMG recording and signal processing for studies carrying out neurophysiological assessments.

| Study | PREPARATION/RECORDING | | | SIGNAL PROCESSING | | | RESULTS |
|---|---|---|---|---|---|---|---|
| | Muscles [Preparation Described] | Recording Device [Sampling Frequency] | Filter Passband [Stim artefact filtering] | Rectification | Cycle averaging | Amplitude Normalization | Output Presented |
| Dy *et al.*, 2010 [14] | Sol, MG, TA, med hams, VL | Hard wired A/D board and customized labVIEW software | 20-1000Hz for resting and standing | P2P amplitude for resting standing. Full wave rectified and peak for stepping. | 12 MMR responses | Mean MMR for each muscle was normalized to sol responses as stim electrode placement was determined by optimization of sol response | Quantitative comparison between NI and SCI for MMR resting and standing, and phase-dependant MMR during stepping |
| | [✓] | [200Hz] | 40-500Hz for stepping | | | | |
| | | | [✓] | | | | |
| Emeliannikov *et al.*, 2016 [15] | RF, BF, TA, and LG | Viasys Viking Select | - | P2P amplitude | - | - | Comparison of MMR, H-Reflex and M-Wave at rest. |
| | [✗] | [NS] | [✗] | | | | |
| Hofstoetter *et al.*, 2018 [16] | RF, BF, TA and TS | DasyLab 11.0 | 1. 10-500Hz | P2P amplitude | - | - | Quantitative comparison of 1st and 2nd MMR amplitude for TSS and ESS. Onset offset and duration of 1st and 2nd MMR responses. Normalized response thresholds for TSS and ESS |
| | [✓] | Codas ADC system | 2. 30-700Hz with add. 500Hz low-pass | | | | |
| | | [2048 and 2002Hz] | [✓] | | | | |
| Hofstoetter *et al.*, 2019 [17] | RF, BF, TA and Sol | Phoenix multichannel EMG system | 10-1000Hz | P2P amplitude | 10 | Response amplitude of 2nd stimulus in each pair was normalized to the respective 1st for increasing inter-pulse interval (20-5000ms) | Quantitative comparison between NI and SCI for recovery of 2nd PRM as inter-pulse interval increased. |
| | [✓] | [2048Hz] | [✓] | | | | |
| Murray and Knikou, 2019 [20] | Sol, MG, PL, TA, med hams, lat ham, RF, and GRC | 1401 Plus System | 10-1000Hz | Full wave rectified AUC for each TEP response. | 15 | Responses at increasing intensities were normalized to the associated max response for recruitment curve. | Quantitative comparison of recruitment curve sigmoid parameters, PAD and HD, before and after 60min TSS. |
| | [✗] | [2000Hz] | [✗] | | | Responses at increasing frequency (0.1–1.0 Hz) normalized to response at 0.1Hz for HD. | |
| | | | | | | Response amplitude of 2nd stimulus of a pair was normalized to the respective 1st for PAD | |
| Atkinson *et al.*, 2020 [26] | RF, VL, med ham, TA, MG, Sol. | MA300 EMG System | - | P2P amplitude | 10 | Recruitment responses normalized to P2P amplitude at the maximum rate of recruitment (RRmax) within each muscle. CTI: 2nd stimulus in each pair was normalized to the respective 1st for increasing inter-pulse interval (40-160ms) | Quantitative comparison of interlimb conditioning between NI and SCI. |
| | [✗] | [5000Hz] | [✗] | | | | |

*(Continued)*

**Table 9.** (Continued)

| Study | PREPARATION/RECORDING | | | SIGNAL PROCESSING | | | RESULTS |
|---|---|---|---|---|---|---|---|
| | Muscles [Preparation Described] | Recording Device [Sampling Frequency] | Filter Passband [Stim artefact filtering] | Rectification | Cycle averaging | Amplitude Normalization | Output Presented |
| Militskova *et al.*, 2020 [25] | RF, med ham, TA, sol | Neuro MEP- (Neurosoft, Ivanovo, Russia) | - | - | 10 | - | Quantitative comparison of SEP latency, threshold and amplitude across (A) 3 stim sites, (B) lying supine vs. standing and (C) pre- post-rehab |
| | [✗] | [5000Hz] | [✗] | | | | |
| Wu *et al.*, 2020 [23] | APB, ADM, FCR, BB | Customized LabVIEW software (National Instruments USB-6363) | 15-2000Hz | - | 10 | Response amplitude of 2nd stimulus of a pair was normalized to the respective 1st (PAD) | Quantitative comparison recruitment curves across stim configuration. Quantitative comparison of PAD across stim intensity between NI and SCI. |
| | [✗] | [5000Hz] | [✗] | | | | |
| Islam *et al.*, 2021 [72] | Sol, MG, TA, PL [✓] | Data acquisition card (National Instruments, Austin, TX, USA) [2000Hz] | 20–500 Hz [✗] | Full wave rectification, linear enveloping via 20Hz low pass filter. Average RMS for entire step cycle. | - | Maximal stepping EMG without stimulation | Quantitative comparison between stim off, stim on (pulse train or single pulses) and post stim of RMS EMG amplitude, mean power frequency, inter/intra limb coordination and h-reflex modulation during stepping in individuals with and without SCI |
| | | | | P2P amplitude for h-reflex | | M-wave for h-reflex | |

Abbreviations: ADM; abductor digiti minimi, APB; abductor pollicis brevis, BB; biceps brachii, AUC; area under curve, BR; brachioradialis, CTI; conditioning-test interval, Delt; deltoid, DGO; ED; extensor digitorum, FCR; flexor carpi radialis, FD; flexor digitorum, GRC; gracilis, ham; hamstrings, HD; homosynaptic depression, lat ham; lateral hamstrings, LG; lateral gastrocnemius, med ham; medial hamstrings, MG; medial gastrocnemius, MMR; multisegmental monosynaptic response, P2P; peak-to-peak, PAD; post-activation depression, PL; peroneus longus, Q; quadriceps, RA; rectus abdominis, RF; rectus femoris, RMS; root mean square, Sol; soleus, SEP; spinally evoked potential, TA; Tibialis Anterior, TB; triceps brachii, TFL; tensor fascia lata, TP; tibialis posterior, TS; triceps surae/calf, VL; vastus lateralis.

*Preparation described refers to a clear description of preparation of the skin before surface electrode application, recording electrode type, orientation, shape and composition as well as interelectrode distance.

†Artifact filtering refers to an attempt made by the authors to account for and remove artifacts contaminating or obscuring the recorded EMG signals such as with the use of a filter.

stimulated and using this data to then inverse filter surface EMG signal channels [16, 18, 19]. The most popular methods for EMG amplitude processing were the use of full-wave rectification [20, 37, 42], the root mean square [6, 19, 68] and integrated EMG value [36, 41, 43, 69]. Several studies chose only to present raw dynamic EMG data [6, 40, 67].

Only a select number of studies normalized EMG amplitude, four of which were therapeutic investigations [18, 42, 43, 70] and seven of which were neurophysiological investigations [14, 16, 17, 20, 23, 26, 72]. Evoked responses were typically normalized to maximal response at a specific stimulus intensity [20, 26, 42] or when evaluating PAD, the amplitude of the second stimulus of a pair was normalized relative the first [17, 20, 23]. A single investigation recording evoked potentials [14], temporally normalized the responses of pulsed stimulus via a foot switch in an attempt to evaluate if the spinal cord could modulate the evoked responses based on the phase of gait. Using a similar methodology, a more recent paper [72] examined the

**Table 10. A summary of dynamic surface EMG data collection, recording and signal processing.** EMG recording and signal processing for therapeutic studies.

| Study | PREPARATION/RECORDING | | SIGNAL PROCESSING | | | | RESULTS |
|---|---|---|---|---|---|---|---|
| | Muscles [Preparation Described] | Recording Device [Sampling Frequency] | Filter Passband [Stim artefact filtration] | Rectification | Cycle averaging | Amplitude Normalization | Output Presented |
| Hofstoetter et al., 2013 [6] | Q, Ham, TA, TS | Wired EMS Handels system | 10–500 Hz | Raw EMG | - | - | Exemplary raw EMG traces during stepping. Qualitative comparison of stim on/off |
| | [✓] | [2048Hz] | [✗] | | | | |
| Gerasimenko et al., 2015 [67] | Sol, MG, TA, med ham, VL | Wired A/D board and customized labVIEW software | 10–10,000Hz | - | - | - | Exemplary raw EMG during voluntary movement. Scatter-plot of antagonistic muscle activity patterns. Qualitative description of EMG change during stim. |
| | [✗] | [10,000Hz] | [✓] | | | | |
| Hofstoetter et al., 2015 [18] | Q, Ham, TA, TS | EMS-Handels system | 10–500 Hz | RMS | 10 gait cycles | EMG during stance and swing phase normalized to muscle activity with stim off | Exemplary raw EMG during stepping. Radar chart of RMS during stance and swing. Qualitative comparison of stim on/off |
| | [✓] | [2048Hz] | [✓] | | | | |
| Bedi, 2016 [68] | Q, Ham, TA, TP | Neurostim Medicad System | 10–500 Hz | RMS | 3 repetitions per side | - | Tables of RMS data during voluntary movement before and after stim |
| | [✓] | [NS] | [✗] | | | | |
| Minassian et al., 2016 [19] | Q, Ham, TA, TS, TFL | Wired Poly-EMG system (EMS-Handels) | 20–500 Hz | RMS | 10 gait cycles | - | Within subject quantitative comparison (ANOVA) of RMS data for treadmill speed x hip extension. Exemplary raw EMG during stepping and standing. |
| | [✓] | [2048Hz] | [✓] | | | | |
| Gad et al., 2017 [41] | Sol, MG, TA, med ham, VL | Wired Powerlab and LabChart. Delsys EMG System also mentioned. | 10–1000 Hz | Integrated | 30 steps | - | Exemplary raw EMG and iEMG during stepping and voluntary movement. Exemplary evoked responses. Qualitative comparison of EMG during passive and active stepping during the intervention. |
| | [✓] | [10,000Hz] | [✓] | | 5 evoked responses | | |
| Freyvert et al,. 2018 [44] | FD, ED, BR, BB, TB | Konigsberg EMG system | - | - | 9 x 3.5 sec hand-grip Repetition | - | Quantitative comparison of raw EMG amplitude across each test phase Exemplary raw EMG during voluntary hand grip tasks. |
| | [✗] | [NS] | [✗] | | | | |
| Rath et al., 2018 [37] | RA, Obl, E-T7, E-L3, RF, delt | Wired Powerlab 16/ 35 | 10–2000 Hz | Full wave rectification | - | - | Exemplary rectified EMG during trunk movement. Quantitative comparison of mean EMG with stim on/off. |
| | [✗] | [2000Hz] | [✓] | | | | |
| Gad et al., 2018 [43] | BB, FD, ED | Wired Powerlab | 10–10,000Hz, 60Hz Notch | Integrated | 5 evoked responses per intensity | Evoked responses normalized to baseline. | Exemplary raw and iEMG of evoked responses. Exemplary raw EMG traces during voluntary contractions. Quantitative comparison of iEMG response to stim on/off and pre- post-intervention |
| | [✗] | [10,000Hz] | [✗] | | | | |
| Inanici et al., 2018 [36] | Delt, TB, BB, BR, ED, FD, ADM, thenar muscles | Wired Delsys Bagnoli system | - | Raw EMG | - | - | Examplary trace of evoked response in thenar muscles compared every 2 weeks across intervention. |
| | [✗] | [1000Hz] | [✗] | | | | |

(Continued)

**Table 10.** (Continued)

| Study | PREPARATION/RECORDING | | SIGNAL PROCESSING | | | | RESULTS |
|---|---|---|---|---|---|---|---|
| | Muscles [Preparation Described] | Recording Device [Sampling Frequency] | Filter Passband [Stim artefact filtration] | Rectification | Cycle averaging | Amplitude Normalization | Output Presented |
| Sayenko et al., 2019 [42] | Sol, TA, VL, med ham | Wired Powerlab | 10–2000 Hz | Full wave rectification and RMS | 6 evoked responses | EMG during anterior/ posterior weight shift normalized to activity during initial position | Exemplary raw EMG during transition and standing. Quantitative comparison of mean EMG between stim on/off, sitting and standing and Quantitative comparison of RMS EMG and motor thresholds across training sessions. |
| | [✗] | [4000Hz] | [✓] | | | | |
| Al'Joboori et al., 2020 [69] | Q, TA, ham, gastric | Wired CED Power 1401 | 10–200 Hz | RMS (0.4s window). Integrated to average level of rectified signal. | 3 trials with flex/ext phases | - | Intergrated EMG activity pre and post trial temporally normalized to phase of movement |
| | [✗] | [2000Hz] | [✗] | | | | |
| Alam et al., 2020 [40] | Q, TA, ham, gastric | Wireless, BTS Telemg | [✗] | P2P amplitude of MEPs. | - | - | Exemplary raw EMG during sit-to-stand and stand-to-sit task. Intensity response curves for each muscle. |
| | [✗] | [2000Hz] | | | | | |
| Meyer et al. 2020 [70] | 1) RF, VM, ST, TA, MG | 1) Wireless Aktos System | 1) 10–500 Hz | Noise corrected RMS | 1) 20 cycles of each rep ankle movements | Tonic stim responses normalized to the respective values in the tSCS-off condition using z-scores | Exemplary data comparing stim off vs. on - 15Hz/30Hz/ 50Hz |
| | | [2000Hz] | 2) 20 Hz— 3000 Hz | | | | normalized RMS stim off vs. on - 15Hz/30Hz/50Hz |
| | 2) Spinal reflex protocol: bilateral TA | 2) Dantec Keypoint Focus Workstation | [✓] | | 2) 5 spinal reflexes | | phase of gait muscle activity |
| | [✗] | [6000Hz] | | | | | |
| Shapkova et al., 2020 [38] | RF, BF, GL, TA | Viasys Viking Select | - | - | - | - | Exemplary trace of H-Reflex and MMR reseponse. |
| | [✗] | [2000Hz] | [✗] | | | | |
| Zhang et al., 2020 [71] | Trap, delt, BB, TB, BR, ECU, FCR, ECR | Wired MA-400 Motion Lab Systems | 10–1000 Hz 60Hz Notch | Integrated | 3 evoked responses | - | Evoked potential amplitudes pre- and post-stimulation intervention |
| | [✗] | [10200 Hz] | [✗] | | | | |

**Abbreviations:** BB; biceps brachii, BR; brachioradialis, delt; deltoid, ED; extensor digitorum, E-L3; erector spinae at level of L3, EMG; electromyography, E-T7; erector spinae at level of T7, ECU; extensor carpi ulnaris, ECR; extensor carpi radialis, FCU; flexor carpi ulnaris, FD; flexor digitorum, Ham; hamstrings, iEMG; integrated EMG, med ham; medial hamstrings, MG; medial gastrocnemius, obl; external oblique, P2P; peak-to-peak, PL; Peroneus Longus, Q; quadriceps, RA; rectus abdominis, RF; rectus femoris, RMS; root mean square, sol; soleus, ST; semitendinosus, TA; Tibialis Anterior, TB; triceps brachii, TFL; tensor fascia lata, TP; tibialis posterior, Trap; trapezius, TB; triceps brachii, TS; triceps surae, VL; vastus lateralis.

* Recording electrodes described refers to a clear description of preparation of the skin before surface electrode application, recording electrode type, orientation, shape and composition as well as interelectrode distance.

† Artifact filtering refers to an attempt made by the authors to account for and remove artifacts contaminating or obscuring the recorded EMG signals such as with the use of a filter or reference EMG electrodes for artefact cancellation.

impact of tSCS on phase dependant modulation of the H-Reflex during gait. The majority of therapeutic studies recorded dynamic EMG during voluntary movements. However, these signals remained for the most part un-normalized, often with the presentation of exemplary un-rectified EMG traces.

### Safety and adverse events

Of all 25 studies included in this review, only 7 explicitly reported on the presence or absence of adverse events [20, 23, 36, 42, 43, 69, 71]. While some studies made comments on stimulation tolerability and pain levels [18, 38, 40, 44], there were insufficient details to rule out all potential safety issues or complications. Four studies reported the complete absence of adverse events while monitoring vital signs throughout, [20, 36, 43, 71]. The associated recorded events from the other studies included: a modest increase in tone in the 24hrs post treatment, unintentional activation of the micturition reflex and voiding during standing, skin breakage and transient redness [42], as well as discomfort during stimulation at high intensities, asymptomatic variations to heart rate and blood pressure and mild side effects possibly related to cervical stimulation including incidents of light headedness, feeling flushed, nausea, a metallic taste, a sensation of 'sharp' breathing, neck pain, and throat discomfort [23]. None of the adverse events recorded were reported to be consistent across treatment sessions, serious, or long-lasting.

## Discussion

### Summary of findings

This review separated studies utilising tSCS into two broad categories: studies evaluating neurophysiological properties of stimulation at a spinal level and those using tSCS as a therapeutic modality for motor recovery. While publications in both categories have grown in number, the quality of the current evidence base is limited, and a large degree of methodological heterogeneity exists between studies. In particular, extensive variability in stimulation parameters and inconsistent processing and/or presentation of EMG signals make it difficult to draw meaningful conclusions about the effects of tSCS on motor engagement. Efforts should be made in future studies to standardise reporting of muscle activity as well as the electrical parameters of tSCS being administered including electrode dimensions and location, charge polarity, phase duration and stimulation frequency.

### A comparison between neurophysiological and therapeutic investigations

Thus far, neurophysiological investigations have focused on the production of evoked motor potentials and the properties of these responses, such as factors affecting response modulation [14, 25, 26] and the characteristics and reflex contributions of stimulation responses [16, 17, 20]. In neurophysiological investigations, stimulation is generally applied with individual or paired pulses at low frequencies to evaluate an evoked response or attain a motor threshold. As such, eliciting specific PRM reflexes is the likely target of these investigations and the stimulation parameters are selected accordingly. Indeed, all neurophysiological investigations recorded the reflex origin of evoked responses.

In contrast, therapeutic investigations aim to exploit tSCS in order to neuromodulate the spinal cord and augment motor responses produced by individuals with SCI [46]. As a result, tSCS is generally applied for a longer duration in combination with rehabilitative activities. It is likely that these aims are considered in the selection of stimulation parameters and may explain why several therapeutic studies have chosen multiple stimulation sites [36, 37, 40–43, 67, 71], as opposed to targeting specific sites as is typical in neurophysiological assessments. Only four therapeutic investigations [18, 19, 69, 70] attempted to determine the reflex nature of motor responses generated by tSCS.

### Stimulation parameters

**Electrode configuration.** The electrode configuration (size, polarity, location) plays an important role in the electrical field produced by tSCS, and consequently the structures

targeted. One of the criticisms of tSCS, when compared to the epidural alternative, is failure to create a localised electrical field thereby limiting activation selectivity [16, 73]. This review found a lack of consensus regarding electrode configuration, particularly in the cervical region, and limited rationale for selected parameters throughout.

All included studies either placed the cathode posteriorly over the spinal column or used biphasic current with alternating polarity. Other studies investigating use of tSCS on neurologically intact individuals placed the cathode anteriorly with the anode posterior [6, 16–18, 23, 36, 43], but this has not been tested in subjects with SCI. Studies describe using monophasic current (n = 10), biphasic current (n = 7) or both (n = 3). Biphasic current has been noted to reduce risk of tissue damage [74], and Hofstoetter *et al.* [16] found that evoked responses were initiated by the abrupt change of polarity of the biphasic stimulation pulses. Only one included study by Wu *et al.* [23] reported directly comparing different polarities and found biphasic 2-ms or monophasic 1-ms pulses with the cathode posterior elicited larger responses at lower intensities.

Electrode location varies throughout included studies, with regard to both rostro-caudal and mediolateral alignment. The cathodes were positioned paravertebrally (n = 7) or centrally (n = 16) but no conclusions have been drawn on the effects of these different positions. One study in uninjured individuals demonstrated that lateralisation of motor responses in lower limbs (right/left differentiation) can be achieved from the placement of stimulating electrodes ~2cm laterally from the lumbar spinous process [75].

With regard to the spinal levels selected, activation specificity of muscle responses has previously been demonstrated along the rostro-caudal axis of the lumbosacral enlargement in neurologically intact [76, 77] and injured [25] individuals. One study tested for optimal evoked responses at different spinal levels prior to commencing the experimental protocol in order to account for inter-individual variability [26]. The consideration of participant-specific parameter selection could better account for anatomical variation between individuals [26, 78], such as conus medullaris termination level [79] or injury scar tissue, thereby providing more targeted treatment. Several therapeutic studies found superior effects from stimulating the coccygeal level along with a lumbar stimulation site, however the rationale for stimulating a region overlying the termination of the cauda equina remains unclear. Indeed, Roy *et al.* [80] found, using a paired pulse stimulation test, that spinal reflexes were optimally elicited with tSCS when the cathode was over the upper-lumber vertebrae (L1-L3), and M-waves were optimally elicited with tSCS when the cathode was placed more caudally (L5, S1). If the proposed mechanism for tSCS involves activation of spinal reflex pathways to lower threshold for CPG or voluntary movement, then it would seem important that therapists confirm that the stimulus is transpinally modulated and not just acting as surrogate FES, as may be the case over the coccyx.

For the stimulation of lower limb responses, the anode was placed over the lower abdomen [6, 16–19, 25, 38, 70] and/or the ASIS/iliac crests [14, 20, 26, 40, 41, 67, 72]. This is consistent with previous investigations testing neurologically intact participants [21, 77, 81, 82]. Only one study placed both the cathode and anode in the same plane over the vertebrae [69]. In the cervical region, there was greater variance as anodes were placed superiorly above the sternal notch [23] or inferiorly on the ASIS/iliac crests [36, 41, 44]. Similarly, in cervical tSCS studies of uninjured individuals, anode locations vary between the left acromion [83], upper trapezius and mid clavicle [84], and anterior neck [22]. A previous study investigating the effects of anode position has shown that it is critical for inducing spinal reflexes [85]. Limited human research has further explored the effects of different anode-cathode configurations as a determinant for stimulation outcomes.

**Electrical characteristics.**   The voltage that builds up on a skin electrode during a pulse depends on the charge density, i.e., the accumulated charge divided by the electrode area.

Large electrodes have lower charge density and therefore lower pulse voltages for the same current. Vargas Luna *et al.* [86] define a charge density threshold beyond which electro-osmotic effects become significant in the skin conduction mechanism. This may have implications for skin comfort and irritation. The charge density in the reviewed studies ranged between 1.8 to 53 μC/cm$^2$. Large electrodes also disperse the current in the underlying tissues which may reduce the likelihood of reaching stimulation thresholds at target neurons while increasing the probability of unwanted collateral stimulation.

For sustained trains of pulses lasting several seconds the accumulated direct current, stimulation of pain receptors and heating effect in the skin must be considered. Monophasic pulse trains produce direct current which can give rise to unwanted electrochemical effects at the electrode site leading to skin irritation and even damage. DC levels higher than 0.5 mA/cm$^2$ at the cathode, and 1.0 mA/cm$^2$ at the anode, are potentially harmful [87]. Even for balanced biphasic waveforms, safety standards such as IEC 60601-2-10 require that the user be advised when the skin current density exceeds 2 mA/cm$^2$. Neurologically intact subjects with normal skin sensation will usually find this current density quite uncomfortable, and even further rigour must be taken by researchers when working with participants with impaired sensation. FDA guidance documents advise against power densities greater than 0.25W/cm$^2$ due to the potential heat damage to tissue [88].

The use of carrier frequency in therapeutic studies seems poorly explored or justified at present. The selection of a relatively high frequency carrier waveform in the stimulation pulse is believed to give more efficient signal transfer because the skin-electrode interface impedance has an area-dependent capacitive component which presents less electrical impedance at higher frequency [89–91]. Many studies use a 1 ms pulse with a 10 kHz carrier that has the effect of chopping the pulse into 10 x 50 μs sub-pulses, which reduces the net charge by 50%. It is difficult to compare stimulation effectiveness between studies because of the wide variation in cathode electrode areas used. The included data does not suggest that the current threshold is less when using a carrier, if anything, the opposite may be the case. Meyer *et al.* [69] report a mean current amplitude of 35 mA for a 1 ms biphasic pulse (no carrier) whereas most of the studies using 1 ms monophasic pulses with 10 kHz carriers report much higher peak currents, which may be necessary to compensate for the reduced net charge (Table 7). Modern stimulators are current controlled and automatically adjust their output voltage to ensure the pre-set current is delivered. This ensures electrical losses in the skin do not affect the current that is delivered, perhaps obviating the need for carriers.

**Position of participants.**   Different testing conditions may alter tSCS motor responses and studies have demonstrated that spinally-evoked muscle response amplitudes are facilitated or supressed depending on positional factors and activity phases [5, 14, 92]. A case report by Militskova *et al.* [25] found that spinally evoked response amplitudes were highest in standing, compared to supine, in an individual with SCI. Conversely, a study of 10 healthy participants by Danner *et al.* [93] found that response amplitudes were higher in supine than standing, but that response thresholds were lowest in standing. These studies suggest that results could be affected by position-dependent changes in the electrical field distribution or afferent input altering spinal excitability. Additionally, body position alters the location of the spinal cord within the vertebral canal [94]. Future studies must consider the effects of activity and body positioning and explore conditions that mimic potential clinical scenarios.

## Interpretation of EMG data

The included studies used EMG signals either to quantify the evoked responses at rest, or the level of muscle activity recorded during voluntary movement. Reported methods were

frequently lacking detail, and in some cases not reflective of best practice for the recording, processing, or presentation of surface EMG [55, 95].

In terms of evaluating the magnitude of evoked responses, the majority of studies reported peak-to-peak amplitude of the unrectified EMG signal during a specific time-window after a tSCS pulse was administered. Exceptions to this included Murray and Knikou [20] who quantified the area under the curve of the rectified waveforms, and Dy *et al.* [14] who evaluated phase-dependant modulation of the evoked response during stepping. The reflex nature of evoked responses was most commonly evaluated by quantifying PAD. Other neurophysiological indices calculated from these evoked EMG waveforms included latency and motor threshold.

A notable methodological consideration for EMG recorded during spinal stimulation, and to a greater extent transcutaneously, is the presence of considerable stimulus artefact within the signal. This review identified several filtering approaches including employing high order band-pass filtering of 30-200Hz [67], lower order 6th order Butterworth filters with a passband of 30-1000Hz [41] and the implementation of blanking intervals based on stimulus artefact recorded from trunk musculature [18, 19]. Of note was the detailed description by Rath *et al.* [37] of a multi-stage "linear adaptive filter" process which was subsequently utilised by Sayenko *et al.* [42]. However, the efficacy of one approach relative to another for optimising signal: noise ratio has yet to be fully elucidated. Unfortunately, the majority of therapeutic studies (n = 8) did not explicitly report any attempt at filtering non-physiological noise associated with tSCS. A recent study proposed a novel filtration method, Artefact Component Specific Rejection, in order to address this technical challenge and dissociate the net muscle activity from stimulation artefact [49]. This filtration method was employed with data from stroke survivors and has not yet been utilised in other neurologically impaired populations [42].

A recent consensus statement on EMG signal normalisation highlighted its importance for comparing muscle activity between measurement sessions and/or experimental conditions [55]. Despite its well-documented importance, we found only two therapeutic studies that attempted to normalize dynamic EMG signals between sessions or experimental conditions. Many of the therapeutic studies present un-normalized and/or un-rectified exemplary EMG traces, providing some limited qualitative evidence of motor engagement during gravity neutral leg movements [67], assisted robotic stepping, sit-to-stand movement [40] or voluntary handgrip task [44]. Other studies attempted to statistically compare un-normalized EMG signals recorded intermittently over several months [41, 44]. In either case, no meaningful conclusions regarding the efficacy of tSCS to alter muscle activity in patients with SCI can be drawn from this data. Future studies attempting to examine the effect of tSCS therapy on muscle activity during dynamic movements are recommended to present EMG envelopes which have been normalized to an appropriate reference maxima [55] and averaged across multiple cycles or repetitions. Examples of this approach can be seen in the detailed qualitative [96] and quantitative [97] comparisons of EMG recorded from patients with SCI during stepping in the presence or absence of eSCS.

## Quality of included trials

Research investigating the effects of tSCS is an emerging field that predominantly consists of exploratory clinical trials with small sample sizes, and studies were unsurprisingly found to be of poor-to-fair quality using the D&B Checklist. All studies scored poorly on external validity due to a lack of balanced protocols and reporting on recruitment methods. Research in this field is difficult to extrapolate to the population as a whole, as people with SCI differ markedly,

even within the same clinical classifications [98]. Despite this challenge, some investigations with this population have attempted to employ balanced protocols with respect to variables such as AIS classification [69, 98]. Moreover, studies lacked comprehensive accounts of their recruitment protocols and only 7 detailed eligibility criteria, [20, 23, 38, 42–44, 69]. Regardless of inherent recruitment challenges, greater transparency is needed.

Studies also scored poorly for internal validity and there was limited use of randomisation, blinding or sham stimulation. The use of non-randomised designs is common in studying individuals with chronic conditions such as SCI due to inherent methodological, ethical, and practical considerations [99, 100]. Despite this, two studies did employ randomisation using a crossover design [37, 42]. In only three studies, assessors were blinded to the intervention [38, 42, 44]. Sayenko *et al.* [42] was the only study that attempted to use a placebo in the form of two sham stimulation conditions; one on a different location on the spinal cord that did not project to the motor pools assessed and another designed to give the sensation of stimulation without targeting motor responses. While these forms of sham stimulation may not be completely inert in their effects, it demonstrates the only attempt to account for the potential placebo effect of stimulation.

## Limitations of this review

We acknowledge that this review is subject to several potential limitations. Due to the variance in terminology in this field and the lack of standardised nomenclature, it is possible that relevant studies may have been missed by our search strategy. Additionally, our eligibility criteria included studies using EMG outcomes and therefore other studies detailing the tSCS parameters may have been excluded. Finally, study outcomes were not possible to pool due to the heterogeneity of included experiments, and therefore conclusions regarding the optimal stimulation parameters and study protocols cannot be drawn.

## Recommendations and future directions

To fully exploit the capacity of tSCS to facilitate motor activity, future research must directly explore the effects of different parameters to determine the optimal conditions for desired motor outcomes. Greater justification for the selection of therapeutic stimulation parameters needs to be provided by experiments that bridge the gap in our understanding of parameter optimisation, clinical application, and the mechanisms that promote motor recovery. The quality of future trials would be improved with better reporting of recruitment methods and intervention protocols and with the application of techniques such as randomisation and sham-stimulation. The presence or absence of adverse events must be explicitly detailed to provide a larger evidence base supporting the safety and feasibility. Finally, studies must also include improve the methodological rigour for data collection, processing and reporting in particular of EMG data.

## Conclusions

The results of this systematic review indicate that studies investigating the effects of tSCS interventions for individuals with SCI face both methodological and measurement deficiencies. While initial investigations have improved our understanding of the neurophysiological impact of this technology and demonstrated its feasibility in motor rehabilitation, greater homogeneity in the reporting of stimulation parameters and outcome measurement will be required to pool cumulative outcomes from small sample sizes. A higher quality of studies will be needed to demonstrate conclusive evidence on the standardised application and uses of tSCS.

## Supporting information

**S1 Checklist. PRISMA 2009 checklist.**
(DOCX)

**S1 Appendix. Search strategy and terminology.**
(DOCX)

## Author Contributions

**Conceptualization:** Clare Taylor, Conor McHugh, Conor Minogue, Neil Fleming.

**Data curation:** Clare Taylor, Conor McHugh, Neil Fleming.

**Formal analysis:** Clare Taylor, Conor McHugh, David Mockler, Conor Minogue, Richard B. Reilly, Neil Fleming.

**Funding acquisition:** Conor Minogue, Richard B. Reilly, Neil Fleming.

**Investigation:** Clare Taylor, Conor McHugh, Neil Fleming.

**Methodology:** Clare Taylor, Conor McHugh, David Mockler, Conor Minogue, Neil Fleming.

**Project administration:** Clare Taylor.

**Supervision:** Richard B. Reilly, Neil Fleming.

**Validation:** Conor McHugh, Neil Fleming.

**Writing – original draft:** Clare Taylor.

**Writing – review & editing:** Clare Taylor, Conor McHugh, Conor Minogue, Richard B. Reilly, Neil Fleming.

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
