## [Decision Letter · Decision Letter 0]

28 Jun 2021

PONE-D-21-17243

Transcutaneous spinal cord stimulation and the generation of motor responses in individuals with spinal cord injury: a methodological review

PLOS ONE

Dear Dr. Taylor,

Thank you for submitting your manuscript to PLOS ONE. After careful consideration, we feel that it has merit but does not fully meet PLOS ONE’s publication criteria as it currently stands. Therefore, we invite you to submit a revised version of the manuscript that addresses the points raised during the review process.

When revising your manuscript, it is expecially important that you explain your inclusion criteria (Why only EMG as outocme measures? Why only studie published between 1995 and 2020?). Besides, you should shorten and sharpen your discussion.

We look forward to receiving your revised manuscript.

Kind regards,

Peter Schwenkreis

Academic Editor

PLOS ONE

Journal Requirements:

2. Please include captions for *all* your Supporting Information files at the end of your manuscript, and update any in-text citations to match accordingly. Please see our Supporting Information guidelines for more information: http://journals.plos.org/plosone/s/supporting-information.

Reviewers' comments:

Reviewer's Responses to Questions

**Comments to the Author**

1. Is the manuscript technically sound, and do the data support the conclusions?

Reviewer #1: Partly

Reviewer #2: Yes

2. Has the statistical analysis been performed appropriately and rigorously? 

Reviewer #1: Yes

Reviewer #2: N/A

3. Have the authors made all data underlying the findings in their manuscript fully available?

Reviewer #1: Yes

Reviewer #2: Yes

4. Is the manuscript presented in an intelligible fashion and written in standard English?

Reviewer #1: Yes

Reviewer #2: Yes

5. Review Comments to the Author

Reviewer #1: Thank you very much for considering me as one of the reviewers of this manuscript and thanks for authors’ hard works. I have a number of comments below.

1) I have found a recent review published: Megia Garcia, Alvaro, et al. "Transcutaneous spinal cord stimulation and motor rehabilitation in spinal cord injury: a systematic review." Neurorehabilitation and neural repair 34.1 (2020): 3-12.

Please explain how your review is different and giving more info to readers.

2) Introduction:

2.1 tSCI is not a novel technique regarding “This is a novel modality under the relatively early stages of investigation.”

2.2 Please clarify; a multiple motor pools and auditory commands.

3) Methods:

Why did you include only studies between 1995- June 2020? Then you miss out 39 studies published before 1995. Can you update the data till May 2021?

Recently, interesting studies are published e.g.

- Inanici, Fatma, et al. "Transcutaneous spinal cord stimulation restores hand and arm function after spinal cord injury." IEEE Transactions on Neural Systems and Rehabilitation Engineering 29 (2021): 310-319.

- Hofstoetter, Ursula S., et al. "Transcutaneous Spinal Cord Stimulation Enhances Walking Performance and Reduces Spasticity in Individuals with Multiple Sclerosis." Brain Sciences 11.4 (2021): 472.

- Benavides, Francisco D., et al. "Cortical and subcortical effects of transcutaneous spinal cord stimulation in humans with tetraplegia." Journal of Neuroscience 40.13 (2020): 2633-2643.

4) Discussion

4.1 Discussion part is too long, better to make it precise.

4.2 “Of the 22 included studies, 7 were case reports, 5 were case series, 3 were crossover trials, 6 were quasi-experimental studies (non-equivalent control group or nonrandomised intervention design) and one was a non-randomised control trial.”

Please explain why there is no or very few RCT studies reported between 1995-2020.

4.3 “Due to electrode sites, tSCI cannot target neural tissue for stimulation.” Have you thought about other forms of TENS e.g., interferential current stimulation that can target the neural tissue?

4.4 It is better to have a reference. “FDA guidance documents advise against power densities greater than 0.25W/cm2 due to the potential heat damage to tissue.”

5) Please see minor errors highlighted in the pdf file, especially a few abbreviations (NI, NIL, @).

Reviewer #2: This methodological review article provides a summary of published studies incorporating transcutaneous Spinal Cord Stimulation (tSCS) as an intervention for people living with Spinal Cord Injury (SCI), which have used electromyography (EMG) as an outcome measure. There have been several review articles recently published covering a similar topic, however this article brings together both therapeutic and neurophysiological studies, and provides considerable detail on the stimulation parameters used, which sets it apart from other recently published reviews. The authors found that published studies are of poor-fair quality, with low subject numbers, and stated that, due to the wide range of methodological approaches (including stimulation parameters used), they were unable to make methodological recommendations for future trials. Nevertheless, the review provides a helpful addition to the rapidly growing tSCS literature.

My main concern with the review is the decision to only include trials that had used EMG as an outcome measure. As stated in the limitations section, “other studies detailing the tSCS parameters may have been excluded”. I did not find adequate justification for the exclusion of tSCS trials that did not incorporate EMG measures, and I feel it is important for the authors to explain their thought process, and justify this inclusion criterion.

Further, specific comments are provided below.

Page 5, search strategy line 3: there is a typo “was kept broad to in an attempt to…”

Table 3: There are several undefined abbreviations (e.g. CR, CS, NI)

Table 6: Why is the phase charge for Dy et al. 30 –83µC rather than 25-83 µC? Why is the Phase charge density only calculated for the upper current limits? For Wu et al. I don’t understand how the phase charge 89 µC was calculated.

Table 7: The study by Gad et al. 2015 (57) incorporated a carrier frequency, which is not stated in the Table. Gad et al. 2015 is a conference proceeding, which presents a single case study. The same case study is also presented in the journal publication by Gad et al. (Gad et al. 2017; your ref 41), so I wonder whether the Gad et al. 2015 study should be removed from the review.

Pages 24-25: It would be useful if the authors could provide a clear comparison between the electrical dosage and characteristics of studies that incorporate carrier frequencies, and those that do not. Is the reduction in charge duration generally compensated for by increased current amplitudes in the studies that incorporate the carrier, or is the overall charge density much lower? I think this is an important point because, as stated by the authors, there is no justification for the carrier, and it would be interesting to know whether the (motor and sensory) effects of tSCS are only dependent on charge density. Further detail on the method used to establish current amplitude (i.e. relative to motor/sensory threshold) for studies that do/do not incorporate a carrier, would also be a helpful addition.

Page 40: there is a typo “This is spite of” change to “This is in spite of”

Page 42; there is a typo “by Dannet et al. (80)” change to “by Danner et al. (80)”

Page 43: “Examples of this approach can be seen in the detailed qualitative (84) and quantitative (85) comparisons of EMG recorded from patients with SCI during stepping in the presence or absence of eSCS”. It might be useful for the authors to briefly describe the procedures used by these studies. It should also be noted that stimulus artefacts are a significantly greater problem for tSCS than eSCS studies, due to the substantially higher currents required.

Figure 1 legend: “the two phases having durations of t1 and t1 resp.” should this be t1 and t2?

6. PLOS authors have the option to publish the peer review history of their article (what does this mean?). If published, this will include your full peer review and any attached files.

Reviewer #1: **Yes: **Sam Parittotokkaporn

Reviewer #2: **Yes: **Lynsey Duffell

---

## [Author Response · Author response to Decision Letter 0]

12 Aug 2021

Copy of responses: 

REVIEWER #1

1.1 I have found a recent review published: Megia Garcia, Alvaro, et al. "Transcutaneous spinal cord stimulation and motor rehabilitation in spinal cord injury: a systematic review." Neurorehabilitation and neural repair 34.1 (2020): 3-12. Please explain how your review is different and giving more info to readers.

Response:

When comparing Megía García et al. (2020) to our study, there are a number of key differences. Firstly, Megía García et al. (2020) focuses only on therapeutic investigations of transcutaneous spinal cord stimulation (tSCS) and the efficacy of this technology in motor rehabilitation. In contrast, our study is centred around a technical appraisal all studies using tSCS including neurophysiological investigations. There are 9 neurophysiological investigations included in this review as well as 4 therapeutic investigations that were not included in the original review. Our focus is on assessing the stimulation parameters and presenting a novel attempt to quantify electrical characteristics to allow for a more direct comparison of parameters. Additionally, we provide as focused review of EMG outcomes as it is a commonly employed measure of tSCS output that can be used to evaluate parameter selection in real time. We thank the reviewers for this comment and the narrative has been edited in the manuscript in an attempt to make this clearer to the reader.

1.2 tSCI is not a novel technique regarding “This is a novel modality under the relatively early stages of investigation.”

Response: 

The manuscript has been revised as per this recommendation. 

1.3 Please clarify, a multiple motor pools and auditory commands.

Response:

This sentence has been re-phrased in the manuscript to add clarity to the meaning.

1.4 Why did you include only studies between 1995- June 2020? Then you miss out 39 studies published before 1995. Can you update the data till May 2021?

Response:

Thank you for your enquiry surrounding this. According to our research, spinal cord stimulation for motor responses in individuals with SCI has garnered much attention in the literature since “stepping-like” movements were induced in individuals with chronic complete SCI that had epidural electrodes implanted for pain management purposes (Dimitrijevic et al., 1998). The use of transcutaneous spinal cord stimulation subsequently emerged in 2007 as an alternative, non-invasive form of therapeutic stimulation (Dimitrijevic et al., 2004; Courtine et al., 2007; Minassian et al., 2007). Bearing this in mind, we did not expect to find any relevant articles prior to this time period. However, we have since realised thanks to your comments that this is a less comprehensive form of appraisal. Therefore, we reviewed each of the 39 studies previously excluded based on their publication date and all of them were ultimately re-categorised under different exclusion criteria in the abstract or full-text screening.

In addition, the data has been updated until May 2021 and 3 further studies have been included. Please see new PRISMA flow chart for further details.

1.5 Recently, interesting studies are published e.g.

- Inanici, Fatma, et al. "Transcutaneous spinal cord stimulation restores hand and arm function after spinal cord injury." IEEE Transactions on Neural Systems and Rehabilitation Engineering 29 (2021): 310-319.

- Hofstoetter, Ursula S., et al. "Transcutaneous Spinal Cord Stimulation Enhances Walking Performance and Reduces Spasticity in Individuals with Multiple Sclerosis." Brain Sciences 11.4 (2021): 472.

- Benavides, Francisco D., et al. "Cortical and subcortical effects of transcutaneous spinal cord stimulation in humans with tetraplegia." Journal of Neuroscience 40.13 (2020): 2633-2643.

Response: 

- Inanici et al., 2021. While an interesting study, it did not include EMG data which is our primary outcome for technical analysis (for reasons detailed below).

- Hofstoetter et al., 2021. This study was excluded as the primary outcomes focus on spasticity and as it was carried out in individuals with multiple sclerosis, both of which were exclusion criteria. The parameters selected for spasticity management with tSCS typically vary from those aiming at producing motor outcomes.

-Benavides et al., 2020. This study failed the full-text screen because it did not include a direct measure of the motor output from transcutaneous spinal stimulation. The intervention in this study was indeed tSCS, however, the outcome was cortical and sub-cortical motor evoked potentials representing the effect of tSCS on cortical and subcortical connectivity. No direct EMG data from transcutaneous spinal cord stimulation was recorded.

1.6 Discussion part is too long, better to make it precise.

Response:

We thank the reviewers for this recommendation and have edited the manuscript accordingly. The discussion surrounding other outcome measures was removed and all other areas we shortened. The order was also reorganised. 

1.7 “Of the 22 included studies, 7 were case reports, 5 were case series, 3 were crossover trials, 6 were quasi-experimental studies (non-equivalent control group or nonrandomised intervention design) and one was a non-randomised control trial.” Please explain why there is no or very few RCT studies reported between 1995-2020.

Response:

Unfortunately, the lack of RCTs is a significant limitation of published research in this field thus far. There have only been two randomized crossover trials and no strict randomized controlled trials published in the literature. However, it is noted there are ethical considerations that can make RCTs difficult to accomplish with this population. 

1.8 “Due to electrode sites, tSCI cannot target neural tissue for stimulation.” Have you thought about other forms of TENS e.g., interferential current stimulation that can target the neural tissue?

Response:

We thank the reviewer for this interesting suggestion. This point could be potentially be addressed in for future appraisals but at present we decided to focus on evaluating this type of stimulation in order to narrow the focus of the review. 

1.9 It is better to have a reference. “FDA guidance documents advise against power densities greater than 0.25W/cm2 due to the potential heat damage to tissue.”

Response:

We thank the reviewers for this recommendation and have edited the manuscript accordingly.

1.10 Please see minor errors highlighted in the pdf file, especially a few abbreviations (NI, NIL, @).

Response:

We thank the reviewers for highlighting these minor errors and have edited the manuscript accordingly.

REVIEWER #2

2.1 My main concern with the review is the decision to only include trials that had used EMG as an outcome measure. As stated in the limitations section, “other studies detailing the tSCS parameters may have been excluded”. I did not find adequate justification for the exclusion of tSCS trials that did not incorporate EMG measures, and I feel it is important for the authors to explain their thought process and justify this inclusion criterion.

Response: 

We would like to thank the reviewers for highlighting the lack of justification surrounding the inclusion of only articles that use EMG outcomes. It was an omission and in correcting it, we believe we have improved the quality of the article. We have added the justification to the introduction and main text that hopefully explains this criterion further. Briefly, EMG allows for the real-time assessment of electrical signals representing motor output facilitated by tSCS. This can be beneficial for the assessment of spinal excitability and the optimization of stimulation parameters. It as outcome employed by both therapeutic investigations and neurophysiological investigations that enable the quantification of the neuromuscular component of tSCS output.

2.2 Page 5, search strategy line 3: there is a typo “was kept broad to in an attempt to…”

Response: We thank the reviewer for highlighting this error and have edited the manuscript accordingly.

2.3 Table 3: There are several undefined abbreviations (e.g. CR, CS, NI)

Response: We thank the reviewer for highlighting these errors and have edited the manuscript accordingly.

2.4 Table 6: Why is the phase charge for Dy et al. 30 –83µC rather than 25-83 µC? Why is the Phase charge density only calculated for the upper current limits? For Wu et al. I don’t understand how the phase charge 89 µC was calculated.

Response: -Table 1, Dy et al., does show a range between 24.7 and 83 mA for the SCI subjects and thanks are given for the identification of this typo. 

-The Phase charge density was only provided for the upper limit to simplify the table and as the focus was on safe maximal levels. Max has been included in the table title to clarify this for the reader. 

-In Wu et al., our understanding is that the maximum current was 89 mA, and the maximum phase duration was 1 ms, hence a phase charge of 89 µC. The authors refer to 2 ms biphasic pulses but our understanding is that the duration of each phase was 1 ms. They also refer to a 1 ms monophasic pulse.

2.5 Table 7: The study by Gad et al. 2015 (57) incorporated a carrier frequency, which is not stated in the Table. Gad et al. 2015 is a conference proceeding, which presents a single case study. The same case study is also presented in the journal publication by Gad et al. (Gad et al. 2017; your ref 41), so I wonder whether the Gad et al. 2015 study should be removed from the review.

Response: While it is very likely that the paper by Gad et al. 2015 did, in fact, use a carrier frequency, it is not documented anywhere in their IEEE publication and we agreed that we could only include a review of what was documented. 

Originally, we elected to include Gad 2015 as alternative outcomes and EMG analysis were employed. However, on further consideration and based on its status as a conference proceeding, we have excluded it from the revised manuscript following your recommendation.

2.6 Pages 24-25: It would be useful if the authors could provide a clear comparison between the electrical dosage and characteristics of studies that incorporate carrier frequencies, and those that do not. Is the reduction in charge duration generally compensated for by increased current amplitudes in the studies that incorporate the carrier, or is the overall charge density much lower? I think this is an important point because, as stated by the authors, there is no justification for the carrier, and it would be interesting to know whether the (motor and sensory) effects of tSCS are only dependent on charge density. Further detail on the method used to establish current amplitude (i.e. relative to motor/sensory threshold) for studies that do/do not incorporate a carrier, would also be a helpful addition.

Response: We would like to thank the reviewer for these very interesting questions. In the discussion, the analysis of the carrier frequency and its potential effect on stimulation thresholds has been expanded upon in order to address these points:

The use of a relatively high frequency carrier waveform in the stimulation pulse is believed to give more efficient signal transfer because the skin-electrode interface impedance has an area-dependent capacitive component which presents less electrical impedance at higher frequency. Many studies use a 1 ms pulse with a 10 kHz carrier that has the effect of chopping the pulse into 10 x 50 µS sub-pulses, which reduces the net charge by 50%. It is difficult to compare stimulation effectiveness between studies because of the wide variation in cathode electrode areas used. The data does not suggest that the current threshold is less when using a carrier, if anything the opposite may be the case. Meyer et al report a mean current amplitude of 35 mA for a 1 ms biphasic pulse (no carrier) whereas most of the studies using 1 ms monophasic pulses with 10 kHz carriers report much higher peak currents, which may be necessary to compensate for the reduced net charge. (See Table 7) Modern stimulators are current controlled and automatically adjust their output voltage to ensure the preset current is delivered. This ensures electrical losses in the skin do not affect the current that is delivered perhaps obviating the need for carriers.

 In future, a study directly comparing the effects of stimulation with and without a carrier frequency would likely be beneficial.

2.7 Page 40: there is a typo “This is spite of” change to “This is in spite of”

We thank the reviewer for highlighting this error and have edited the manuscript accordingly

2.8 Page 42; there is a typo “by Dannet et al. (80)” change to “by Danner et al. (80)”

We thank the reviewer for highlighting this error and have edited the manuscript accordingly

2.9 Page 43: “Examples of this approach can be seen in the detailed qualitative (84) and quantitative (85) comparisons of EMG recorded from patients with SCI during stepping in the presence or absence of eSCS”. It might be useful for the authors to briefly describe the procedures used by these studies. It should also be noted that stimulus artefacts are a significantly greater problem for tSCS than eSCS studies, due to the substantially higher currents required.

In order to address this question, the procedures used by these studies have been described in more detail within the sentence immediately preceding this statement. “Future studies attempting to examine the effect of tSCS therapy on muscle activity during dynamic movements are recommended to present EMG envelopes which have been normalized to an appropriate reference maxima (93) and averaged across multiple cycles or repetitions.”

Regarding stimulus artefact, we concur with the reviewers that the problem of stimulus artefact is more significant in tSCS. We have noted this in our discussion and provided additional detail on currently published methods to reduce or eliminate stimulus artefact in tSCS. 

2.10 Figure 1 legend: “the two phases having durations of t1 and t1 resp.” should this be t1 and t2?

We thank the reviewer for highlighting this error and have edited the manuscript accordingly

---

## [Decision Letter · Decision Letter 1]

1 Sep 2021

PONE-D-21-17243R1

Transcutaneous spinal cord stimulation and motor responses in individuals with spinal cord injury: a methodological review.

PLOS ONE

Dear Dr. Taylor,

Thank you for submitting your manuscript to PLOS ONE. After careful consideration, we feel that it has merit but does not fully meet PLOS ONE’s publication criteria as it currently stands. Therefore, we invite you to submit a revised version of the manuscript that addresses the points raised during the review process.

We look forward to receiving your revised manuscript.

Kind regards,

Peter Schwenkreis

Academic Editor

PLOS ONE

Journal Requirements:

Reviewers' comments:

Reviewer's Responses to Questions

**Comments to the Author**

1. If the authors have adequately addressed your comments raised in a previous round of review and you feel that this manuscript is now acceptable for publication, you may indicate that here to bypass the “Comments to the Author” section, enter your conflict of interest statement in the “Confidential to Editor” section, and submit your "Accept" recommendation.

Reviewer #1: All comments have been addressed

Reviewer #2: (No Response)

2. Is the manuscript technically sound, and do the data support the conclusions?

Reviewer #1: Yes

Reviewer #2: Yes

3. Has the statistical analysis been performed appropriately and rigorously? 

Reviewer #1: Yes

Reviewer #2: Yes

4. Have the authors made all data underlying the findings in their manuscript fully available?

Reviewer #1: Yes

Reviewer #2: Yes

5. Is the manuscript presented in an intelligible fashion and written in standard English?

Reviewer #1: Yes

Reviewer #2: Yes

6. Review Comments to the Author

Reviewer #1: Authors have addressed all reviewers' comments appropriately. This revised manuscript is recommended for publication.

Reviewer #2: The authors have addressed all of my comments, with thanks. I have a couple of further (minor) comments following their replies:

1) It would be helpful if the authors are able to provide a reference for the statement "The use of a relatively high frequency carrier waveform in the stimulation pulse is believed to give more efficient signal transfer because the skin-electrode interface impedance has an area-dependent capacitive component which presents less electrical impedance at higher frequency".

2) Given the extended period over which the review was performed, there is an additional article, which I believe should have been included:

Al’joboori, Y., Massey, S.J., Knight, S.L., Donaldson, N. & Duffell, L.D. The Effects of Adding Transcutaneous Spinal Cord Stimulation (tSCS) to Sit-To-Stand Training in People with Spinal Cord Injury: A Pilot Study. Journal of Clinical Medicine, 2020. 9(9): p. 2765.

7. PLOS authors have the option to publish the peer review history of their article (what does this mean?). If published, this will include your full peer review and any attached files.

Reviewer #1: **Yes: **Sam Parittotokkaporn

Reviewer #2: **Yes: **Lynsey Duffell

---

## [Author Response · Author response to Decision Letter 1]

2 Nov 2021

Thank you for your critical appraisal of this journal submission. Edits have been made to the manuscript in an attempt to appropriately address these suggestions. Author comments are outlined below.

Comments to the Author

1) It would be helpful if the authors are able to provide a reference for the statement "The use of a relatively high frequency carrier waveform in the stimulation pulse is believed to give more efficient signal transfer because the skin-electrode interface impedance has an area-dependent capacitive component which presents less electrical impedance at higher frequency".

Response: 

Thank you for highlighting this area as requiring referencing. We have added two references to this statement that detail the capacitance of the skin barrier and the effectiveness of the high frequency carrier. 

2) Given the extended period over which the review was performed, there is an additional article, which I believe should have been included:

Al’joboori, Y., Massey, S.J., Knight, S.L., Donaldson, N. & Duffell, L.D. The Effects of Adding Transcutaneous Spinal Cord Stimulation (tSCS) to Sit-To-Stand Training in People with Spinal Cord Injury: A Pilot Study. Journal of Clinical Medicine, 2020. 9(9): p. 2765.

Response: 

Thank you for bringing this article to our attention. Our librarian believes it to have been missed by the search MeSH due to the keywords we included surrounding EMG. This article is now included in the review paper.

---

## [Editor Report · Decision Letter 2]

4 Nov 2021

Transcutaneous spinal cord stimulation and motor responses in individuals with spinal cord injury: a methodological review.

PONE-D-21-17243R2

Dear Dr. Taylor,

We’re pleased to inform you that your manuscript has been judged scientifically suitable for publication and will be formally accepted for publication once it meets all outstanding technical requirements.

Kind regards,

Peter Schwenkreis

Academic Editor

PLOS ONE
---

## [Editor Report · Acceptance letter]

8 Nov 2021

PONE-D-21-17243R2 

Transcutaneous spinal cord stimulation and motor responses in individuals with spinal cord injury: a methodological review 

Dear Dr. Taylor:

I'm pleased to inform you that your manuscript has been deemed suitable for publication in PLOS ONE. Congratulations! Your manuscript is now with our production department. 

Kind regards, 

on behalf of

Dr. Peter Schwenkreis 

Academic Editor

PLOS ONE